



# Optimal Disturbances of Blocking: A Barotropic View

Bin Shi[1,3], Dehai Luo[2,4], and Wenqi Zhang[2]

[1]Academy of Mathematics and Systems Science, Chinese Academy of Sciences, Beijing 100190, China
[2]Institute of Atmospheric Physics, Chinese Academy of Sciences, Beijing 100029, China
[3]School of Mathematical Sciences, University of Chinese Academy of Sciences, Beijing 100049, China
[4]College of Earth and Planetary Sciences, University of Chinese Academy of Sciences, Beijing 100049, 6 China

**Correspondence:** Bin Shi (shibin@lsec.cc.ac.cn)

**Abstract.** In this paper, we explore optimal disturbances of blockings in the equivalent barotropic atmosphere using the conditional nonlinear optimal perturbation (CNOP) approach. Considering the initial blocking amplitude, the optimal disturbance exhibits a solitary wave-like pattern. As the size increases incrementally, the spatial pattern becomes more concentrated, and the nonlinear evolution becomes more pronounced. During the evolution, it only focuses on gradually intensifying the blocking amplitude without any other influence. Additionally, based on the medium-range experiments, the time-delay optimal disturbance appears to lead to larger errors, making it more challenging to predict. Considering the preexisting synoptic-scale eddies, the optimal disturbance displays a sharply concentrated pattern, even more concentrated by increasing the size. However, it is worth noting that the nonlinear evolution undergoes significant changes, compared to disturbances of the initial blocking amplitude. Meanwhile, we find that the optimal disturbance not only strongly impacts the amplitude of blockings but also their shape, making eddy straining and wave breaking more chaotic and predominant, further influencing the development of weather extremes. This suggests that blockings are more sensitive to perturbations of preexisting synoptic-scale eddies than initial blocking amplitudes. Furthermore, the perturbations of the synoptic-scale eddies are more likely to lead to the development of weather extremes, making them less predictable. In medium-range experiments, it is also found that time-delay disturbances result in larger errors, particularly during the decay period. Finally, we discuss how the variations of westerly wind influence optimal disturbances in spatial patterns and nonlinear evolution as well as their relation to predictability.

## 1 Introduction

Weather extremes have a significant impact on society as they can pose a threat to human life and safety, as well as cause significant economic damage and disruption. For instance, heat waves and extreme droughts can lead to devastating forest fires, damaging agriculture and causing air pollution that poses health risks (Witte et al., 2011). Similarly, cold spells with low temperatures and heavy snowfall can greatly disrupt transportation systems and daily life (Davolio et al., 2015). Additionally, floods caused by heavy precipitation are another type of high-impact weather event that can result in severe consequences, affecting infrastructure, displacing communities, and causing property damage (Lenggenhager et al., 2019). Despite the diverse nature of extreme weather events, they do share a common factor — the prevailing large-scale flow pattern in the troposphere over the North Pacific and North Atlantic Oceans, which is strongly influenced by atmospheric blocking (hereinafter referred



to as *blocking*). Hence, understanding the mechanism behind blocking is crucially important for comprehending and predicting these high-impact weather events (Kautz et al., 2022).

Before delving into the mechanism study of blockings, it is important to understand their features. These blockings are characterized as long-lasting, quasi-stationary, and self-preserving in the midlatitudes, as highlighted in (Liu, 1994; Nakamura and Huang, 2018). In the unfiltered geopotential height field, blocking flow often manifests as a significant meandering of westerly jet streams, as described in (Berggren et al., 1949). This meandering resembles the westerly winds flowing around and bypassing the obstacle created by the blocking pattern. According to the pioneering study (Rex, 1950), a key feature of blocking is the abrupt transition from a zonal (east-west) to a meridional (north-south) flow pattern. This transition often leads to the splitting of the jet stream into two branches around the blocking. Generally, these blockings can be classified into three types, dipole blockings (Hoskins et al., 1985; Pelly and Hoskins, 2003; Weijenborg et al., 2012; Masato et al., 2013), Omega blockings (Häkkinen et al., 2014; Steinfeld and Pfahl, 2019), and amplified ridges (Sousa et al., 2018), with more examples provided in (Woollings et al., 2018). Earlier studies, such as (Berggren et al., 1949; Charney and DeVore, 1979; Tung and Lindzen, 1979; Shutts, 1983; Illari and Marshall, 1983; Holopainen and Fortelius, 1987; Mullen, 1987), suggested that blockings were primarily caused by traveling synoptic-scale eddies and large-scale topography. However, Ji and Tibaldi (1983) conducted numerical experiments, indicating that the topographic forcing plays a secondary role compared to traveling synoptic-scale eddies. Furthermore, the observation that dipole blockings mainly occur downstream of the storm track in the Pacific or Atlantic basin supports the idea that synoptic-scale eddies likely contribute to the formation and maintenance of dipole blocking downstream of the storm tracks (Illari and Marshall, 1983; Holopainen and Fortelius, 1987; Mullen, 1987; Nakamura and Wallace, 1993).

There have been several classical and theoretical models put forward to explain the mechanisms behind the maintenance of blocking patterns. Three well–known models in this regard are the global theory of multiple flow equilibria proposed in (Charney and DeVore, 1979), the local theories of modon proposed in (McWilliams, 1980), and the eddy straining proposed in (Shutts, 1983). In the global theory of multiple flow equilibria, Charney and DeVore (1979) utilized a highly-truncated, nonlinear, barotropic channel model to study the blocking phenomenon from a global perspective. However, observations have indeed shown that most blocking events are primarily a local phenomenon (Dole and Gordon, 1983; Diao et al., 2006). This suggests that a local approach appears to be more consistent with synoptic blocking observations. In the past, McWilliams (1980) utilized modon or vortex pair solutions of the equivalent barotropic vorticity equation as nonlinear free modes to describe the observed blocking features over the Atlantic region. However, it was also noted in (McWilliams, 1980) that the existing condition of the modon solution is not easily satisfied by the observed mean zonal wind. According to (Higgins and Schubert, 1994), the composite field of observed blocking events does not align with the modon or vortex pair structure. Shutts (1983) proposed an eddy straining mechanism that takes into account the time-mean eddy vorticity flux, which aligns with the observed maintenance of blocking and suggests that the eddy straining around the blocking region's two sides plays a crucial role. However, Shutts (1983) also mentioned that blocking is essentially an unsteady phenomenon, meaning that the dynamic life cycle, including onset, growth, maintenance, and decay, is not fully explained by the eddy straining mechanism. Further studies conducted in (Pierrehumbert and Malguzzi, 1984; Haines and Marshall, 1987; Holopainen and Fortelius, 1987) have





expanded on the three well-known theoretical models of blockings. Additionally, Farrell and Ioannou (1996) and Mak and Cai (1989) proposed another viewpoint that suggests barotropic instability as a factor in the occurence of blocking events.

According to (Berggren et al., 1949), the concept of spatial scale separation has emerged, highlighting the interaction between different scales in blocking systems. This interaction involves the interply between fast-moving synoptic-scale eddies and quasi-stationary planetary-scale blocking, which can result in the presence of a low-PV (potential vorticity) air mass. The idea

of the eddy straining mechanism, as described in (Shutts, 1983), further supports this concept by suggesting a "drip-feeding" of low-PV air that replaces the original air mass. Additionally, subsequent wave-breaking events, as described by (Hoskins et al., 1985), serve as another means of replacing the original air mass in this exchange. In contrast to the previous steady-state theorems that primarily focused on the time-mean eddy vorticity flux (Berggren et al., 1949; Shutts, 1983; Pierrehumbert and Malguzzi, 1984; Hoskins et al., 1985; Haines and Marshall, 1987; Holopainen and Fortelius, 1987), recent works by Luo

(2000, 2005) have introduced a dynamic consideration of scale separation in the spatial structure. This approach takes into account the westard propagation of a Rossby wave packet and separates it into a planetary-scale blocking anomaly and preexisting synoptic-scale eddies, based on the background westerly wind. As discussed in (Luo, 2000, 2005), asymptotic analysis has revealed that the slow-varying amplitude of the planetary-scale blocking anomaly behaves like a solitary wave, governed by the forced nonlinear Schrödinger (NLS) equation. Additionally, the preexisting synoptic-scale eddies act as an external force

on the blocking anomaly. Furthermore, Luo et al. (2014) introduced the eddy-blocking matching (EBM) mechanism and the nonlinear multiscale interaction (NMI) model to provide insights into the dynamics of blocking flows. The EBM mechanism helps explain how synoptic-scale eddies can either enhance or suppress a blocking flow. Surprisingly, the NMI model accurately captures the entire life cycle of blockings, including their onset, growth, maintenance, and decay. Additionally, Luo et al. (2019) proposed a theory that elucidates how the meridional gradient of potential vorticity ($PV_y$) influences the dispersive and

nonlinear behavior of blocking. This theory finds support in observations of the background westerly jet stream (Luo et al., 2019).

Despite the progress made in understanding blocking through various theories, accurately numerical predicting the blocking event in weather forecasts remains a challenge (Zhang et al., 2019). The abrupt onset of block flow, as observed in midlatitudes weather over the past century, contributes to the difficulty (Vautard, 1990). This is primarily attributed to the inherent instability

of fluid dynamics, regardless of whether it is in normal or nonnormal modes. In other words, the instability of fluid dynamics act as a barrier that hampers accurate predictions (Pierrehumbert, 1984; Pierrehumbert and Swanson, 1995; Swanson, 2001). The conditional nonlinear optimal perturbation (CNOP) approach, introduced by Mu et al. (2003), is indeed a valuable method for quantifying fluid instability using nonlinear optimization techniques. Unlike approaches that rely on linear approximation assumptions, the CNOP approach takes into account the full nonlinear effects within the system. By maximizing the objective

value while adhering to reasonable physical constraints at a fixed time $T$, the CNOP approach helps identifies the most unstable scenario, or says optimal disturbance. These optimal disturbances, also referred to as optimal precursors, often serve as important signals that can lead to some unstable fluid phenomenon being studied. Therefore, the CNOP approach has been widely applied in various fields such as fluid dynamics, atmospheric science, and oceanography. It has been used to study phenomena like turbulence in shear flows (Pringle and Kerswell, 2010; Kerswell, 2018), disturbance energy in vortex-pairs (Navrose et al.,



2018), detecting blocking onset (Mu and Jiang, 2008), typhoon observations (Mu et al., 2009; Qin and Mu, 2012), predictabil-
ity of El Niño-Southern Oscillation (Duan et al., 2009; Duan and Hu, 2016) and variations in the Kuroshio path (Wang and Mu,
2015). Furthermore, recent advancements in statistical machine learning techniques have further improved and accelerated the
CNOP approach in practical applications (Shi and Sun, 2023; Shi and Ma, 2023).

In this paper, we investigate optimal disturbances of blockings using the NMI model, where the goal is to understand their
static and dynamic behaviors and their relation to predictability. In Section 2, the derivation of the NMI model and the basic
CNOP settings are briefly described to obtain the optimal disturbances. Section 3 provides a theoretical analysis of the opti-
mal disturbance of the initial blocking amplitude, including spatial patterns, nonlinear evolution, the eastward propagation of
blockings, and the time-delay effect. Additionally, a one-by-one comparison is made in Section 3 with the optimal disturbance
of the preexisting synoptic-scale eddies in terms of spatial patterns, nonlinear evolution, the eastward propagation of blockings,
and the time-delay effect. Section 5 discusses how variations in westerly wind influence optimal disturbances in spatial patterns
and nonlinear evolution, as well as their relation to predictability. Finally, in Section 6, this paper concludes with a summary
and discussion.

## 2 The NMI model and optimal disturbances

In this section, we provide a brief description of the barotropic NMI model, which has been derived and developed in (Luo,
2000, 2005; Luo et al., 2014, 2019). The barotropic NMI model serves as a mathematical framework used to study atmospheric
phenomena, particularly those related to blockings. After introducing the barotropic NMI model, we proceed to formalize
the objective functions that need to be maximized. These objective functions play a crucial role in determining the optimal
disturbances of both the initial blocking amplitude and the preexisting synoptic-scale eddies. By maximizing these objective
functions, our aim is to find the optimal perturbations that contribute to the occurrence and development of blockings.

### 2.1 The NMI Model

In the initial stage, we provide a list of various values for the object parameters in Table 1. Let $F = (L/R)^2$ be the Froude
number, where $R \approx L$ is the radius of Rossby deformation. The meridional gradient of the Coriolis parameter at the given
latitude $\varphi_0$ is denoted as $\beta_0$, and the nondimensional parameter is set as $\beta = \beta_0 L^2/U$. Typically, the background westerly wind
is observed to have a speed of approximately $7\,m/s$ (Luo, 2005). Considering that the dimension of wind speed is $10\,m/s$, we
set the nondimensional wind speed as $U = 0.7$.

In the given context, we consider the zonal westerly wind, denoted as $U = U(y)$. Regarding a blocking event, its nondi-
mensional governing equation is expressed as the barotropic quasi-geostrophic equation with $x$-periodic and lateral boundary





| object | parameter | value |
|---|---|---|
|  | reference latitude | $\varphi_0 = 55°N$ |
| horizontal scale | characteristic length | $L \sim 10^6 m$ |
|  | characteristic wind speed | $U \sim 10 ms^{-1}$ |
| $\beta$-channel | nondimensional width | $L_y = 5$ |
| Total Rossby wave packet | nondimensional zonal wavenumber | $k_0 = 1/(6.371 \cos \varphi_0)$ |
|  | nondimensional wind speed (uniform background westerly) | $U = 0.7$ |
| blocking dipole | nondimensional zonal wavenumber | $k = 2k_0$ |
| preexisting synoptic eddies | nondimensional zonal wavenumber | $k_1 = 9k_0$ |
|  | nondimensional zonal wavenumber | $k_2 = 11k_0$ |
|  | zonal location | $x_T = 1.435$ |
|  | amplitude | $a_0 = 0.17$ |
|  | variance parameter | $\mu = 1.2$ |
|  | variance parameter | $\epsilon = 0.24$ |

**Table 1.** The values of the object parameters in the NMI model.

conditions:

$$\frac{\partial}{\partial t}\left(\nabla^2 \psi_T - F\psi_T\right) + J(\psi_T, \nabla^2 \psi_T) + \beta \frac{\partial \psi_T}{\partial x} = 0, \tag{1a}$$

$x -$ periodic : $\quad \psi_T(-L_x, y, t) = \psi_T(L_x, y, t),$ (1b)

$y -$ lateral : $\quad \left.\frac{1}{2L_x}\int_{-L_x}^{L_x} \frac{\partial \psi_T}{\partial y} dx\right|_{y=0} = -U(0), \quad \left.\frac{1}{2L_x}\int_{-L_x}^{L_x} \frac{\partial \psi_T}{\partial y} dx\right|_{y=L_y} = -U(L_y),$ (1c)

where $\psi_T$ is the instantaneous total streamfunction. Then, we can decompose the total streamfunction $\psi_T$ by scales into three parts as

$$\psi_T = \overline{\psi} + \psi + \psi', \tag{2}$$

where $\overline{\psi} = \overline{\psi}(y) = -\int_0^y U(y')dy'$ represents the basic westerly flow, which is only dependent on the meridional direction $y$, $\psi = \psi(x,y,t)$ represents the planetary-scale blocking anomaly, and $\psi' = \psi'(x,y,t)$ represents the preexisting synoptic-scale eddies. Based on observations in the mid-latitudes of the northern hemisphere (Colucci et al., 1981), the planetary-scale blocking anomaly $\psi$ in the zonal direction exhibits a single wave with wavenumber $k = 2k_0$ (Table 1), assuming a corresponding frequency of $\omega$ as discussed in (Charney and DeVore, 1979; Luo, 2005). In the case of the synoptic-scale eddies in the zonal direction, it is believed that they are a superposition of two single waves with wavenumbers, $k_1 = 9k_0$ and $k_2 = 11k_0$ (Ta-





ble 1). The corresponding frequencies for these waves are $\omega_1$ and $\omega_2$, respectively, as mentioned in (Luo, 2005; Luo et al., 2007). Regarding the equivalent barotropic atmosphere, it is widely recognized that the potential vorticity ($PV$) is governed by the equation $PV = f_0 + \beta y - U_y - F\overline{\psi}$, as described in (Pedlosky, 1987). By substituting the instantaneous total stream-function (2) into the nondimensional barotropic quasi-geostrophic equation, (1a), we can establish a relationship between the

three wavenumbers. This relationship leads to two equations for the planetary-scale blocking anomaly $\psi$ and the preexisting synoptic-scale eddies $\psi'$ as

$$\left(\frac{\partial}{\partial t} + U\frac{\partial}{\partial x}\right)\left(\nabla^2\psi - F\psi\right) + J(\psi, \nabla^2\psi) + PV_y\frac{\partial\psi}{\partial x} = -J(\psi', \nabla^2\psi')_P, \tag{3a}$$

$$\left(\frac{\partial}{\partial t} + U\frac{\partial}{\partial x}\right)\left(\nabla^2\psi' - F\psi'\right) + PV_y\frac{\partial\psi'}{\partial x} = -J(\psi', \nabla\psi) - J(\psi, \nabla^2\psi'), \tag{3b}$$

where the meridional gradient of potential vorticity ($PV_y$) satisfies the equation $PV_y = \beta + FU - U_{yy}$, which indicates the $PV_y$ is

slow-varying and the subscript $P$ represents the force driven by the synoptic-scale eddies, which is denoted as $-J(\psi', \nabla^2\psi')$ and has the wavenumber $2k_0$. Indeed, the relative vorticity, represented as $q'$, can be expressed as $q' = \nabla^2\psi' - F\psi'$. By considering this equation, we can derive that the synoptic-scale eddies satisfy $J(\psi', \nabla^2\psi')_P = \nabla \cdot (\mathbf{v}'q')_P$, which represents the planetary-scale component of the divergence of the eddy vorticity flux induced by the preexisting synoptic-scale eddies. It is important to note that the planetary-scale component of the divergence of the eddy vorticity flux $\nabla \cdot (v'q')_P$ is time-dependent. On the

contrary, the time-mean $\nabla \cdot \overline{(v'q')}$ in the eddy straining model is time-independent, as discussed in (Shutts, 1983; Haines and Marshall, 1987).

Recall the classical multiscale decomposition, we can decompose the spatial and temporal components using the parameters $\{X_k = \epsilon^k x\}_{k=0}^{\infty}$ and $\{T_k = \epsilon^k t\}_{k=0}^{\infty}$, respectively. Here, $\epsilon$ is a small parameter and $k$ represents the scale. This decomposition allows us to analyze and understand the behavior of the blocking anomaly and the synoptic-scale eddies at different scales.

Using this decomposition, we can express the wavefunctions of the planetary-scale blocking anomaly and the synoptic-scale eddies as follows:

$$\psi = \psi(x, y, t; X_1, T_1; X_2, T_2; \cdots), \quad \text{and} \quad \psi' = \psi'(x, y, t; X_1, T_1; X_2, T_2; \cdots). \tag{4}$$

Without loss of generality, let us consider the planetary-scale blocking anomaly $\psi$ as an example. By utilizing the multiscale decomposition (4), we can express the temporal and spatial derivatives of the streamfunction $\psi$ as follows:

$$\frac{d\psi}{dt} = \frac{\partial\psi}{\partial t} + \epsilon\frac{\partial\psi}{\partial T_1} + \epsilon^2\frac{\partial\psi}{\partial T_2} + \cdots, \quad \text{and} \quad \frac{d\psi}{dx} = \frac{\partial\psi}{\partial x} + \epsilon\frac{\partial\psi}{\partial X_1} + \epsilon^2\frac{\partial\psi}{\partial X_2} + \cdots. \tag{5}$$

Regarding the streamfunctions of the planetary-scale blocking anomaly and the synoptic-scale eddies (4), we can expand them asymptotically as follows:

$$\psi = \epsilon\psi_1(x, y, t; X_1, T_1; X_2, T_2; \cdots) + \epsilon^2\psi_2(y; X_1, T_1; X_2, T_2; \cdots) + \cdots, \tag{6a}$$

$$\psi' = \epsilon^{\frac{3}{2}}\psi'_1(x, y, t; X_1, T_1; X_2, T_2; \cdots) + \epsilon^{\frac{5}{2}}\psi'_2(x, y, t; X_1, T_1; X_2, T_2; \cdots) + \cdots, \tag{6b}$$

where the fast-varying variable of $\psi_2$ is only meridional or only dependent on $y$, as it represents the blocking's feedback to the zonal-mean westerly wind. Using the derivatives (5) and the asymptotic expansion (6), we employ Wentzel-Kramers-Brillouin (WKB) method from asymptotic analysis (Nayfeh, 2008) to derive the NMI model as:





(1) The nondimensional streamfunctions of the blocking wavy anomaly $\psi_1$, the associated zonal-mean anomaly $\psi_2$ and the preexisting synoptic-scale eddies $\psi_1'$ are represented as follows:

$$\psi_1 = \frac{1}{\epsilon}\sqrt{\frac{2}{L_y}}\left(Be^{i(kx-\omega t)} + \overline{B}e^{-i(kx-\omega t)}\right)\sin\left(my - \frac{\pi}{4}\right), \tag{7a}$$

$$\psi_2 = -\frac{g|B|^2\cos(2my)}{\epsilon^2}, \tag{7b}$$

$$\psi_1' = \frac{2F_0}{\epsilon^{\frac{3}{2}}}\left(\cos(k_1 x - \omega_1 t) - \cos(k_2 x - \omega_2 t)\right)\sin\left(\frac{my}{2} - \frac{\pi}{8}\right), \tag{7c}$$

where $F_0 = a_0 \exp\left[-\mu\epsilon^2(x + x_T)^2\right]$[1] and the parameters are calculated as $m = -2\pi/L_y$ and

$$g = \frac{4mk^2(m^2 + k^2 + F)^2}{PV_y L_y \left[(4m^2 + F)(m^2 + F - k^2) - (m^2 + k^2 + F)^2\right]}.$$

(2) Both the phase and group velocities of the planetary-scale blocking anomaly and the phase velocities of the synoptic-scale eddies are derived separately as

$$c = \frac{\omega}{k} = U - \frac{PV_y}{m^2 + k^2 + F}, \tag{8a}$$

$$c_g = \frac{\partial \omega}{\partial k} = U - \frac{PV_y(m^2 - k^2 + F)}{(m^2 + k^2 + F)^2}, \tag{8b}$$

$$c_1 = \frac{\omega_1}{k_1} = U - \frac{PV_y}{\frac{m^2}{4} + k_1^2 + F}, \qquad c_2 = \frac{\omega_2}{k_2} = U - \frac{PV_y}{\frac{m^2}{4} + k_2^2 + F}. \tag{8c}$$

(3) The blocking amplitude $B$ obeys the 1-dimensional forced NLS equation with the periodic boundary condition as

$$\begin{cases} i\left(\dfrac{\partial B}{\partial t} + c_g \dfrac{\partial B}{\partial x}\right) + \lambda\dfrac{\partial^2 B}{\partial x^2} + \delta|B|^2 B + GF_0^2 \exp(-i\Delta\omega t) = 0, \\ B(0, -L_x) = B(0, L_x), \end{cases} \tag{9}$$

where $\Delta\omega = \omega_2 - \omega_1 - \omega$ and the parameters are set as

$$\begin{cases} \lambda = \dfrac{PV_y k\left[3(m^2 + F) - k^2\right]}{(m^2 + k^2 + F)^3}, \\ \delta = \dfrac{gkm(3m^2 - k^2)}{m^2 + k^2 + F}, \\ G = -\sqrt{\dfrac{L_y}{2}} \cdot \dfrac{m(k_1 + k_2)^2(k_2 - k_1)}{4(m^2 + k^2 + F)}. \end{cases} \tag{10}$$

The detailed derivation of the NMI model is shown in Appendix A.

---

[1]The external force $F_0$, acting as a filter for the waves, indeed serves as the core ingredient of the preexisting synoptic-scale eddies $\psi_1'$. Therefore, unless specifically mentioned afterward, we use the external force $F_0$ to represent the preexisting synoptic-scale eddies.




## 2.2 The basic CNOP settings of optimal disturbances


In the previous derivation of the NMI model, it has been established that the 1-dimensional forced NLS equation, particularly eq. (9), is of great importance in understanding the blocking. Concretely, this equation plays a significant role in describing the dynamic behavior of blocking. The motion of the blocking amplitude in the 1-dimensional forced NLS equation (9) is determined by three factors: the initial blocking amplitude $B_0$, the preexisting synoptic-scale eddies $F_0$ and the background westerly

wind $U$.[2] Traditionally, the Lyapunov exponent has been used to characterize the nonlinear error growth (Lucarini and Gritsun, 2020). However, it is applicable only to finite-dimensional dynamical systems as it requires computing the maximum of finite eigenvalues. Therefore, it does not work for any partial differential equation since it corresponds to an infinite-dimensional dynamical system with an unbounded maximum eigenvalue. It is indeed an interesting question to investigate the effects of perturbations in the initial blocking amplitude $B_0$ and the preexisting synoptic-scale eddies $F_0$ on the motion of blocking. Since

both $B_0$ and $F_0$ are 1-dimensional functions, it raises curiosity about how variations in these parameters affect the evolution behavior of blocking. Additionally, it is worth exploring how changes in the westerly wind $U$ interact with these perturbations to influence the motion of blocking. Understanding these relationships can provide valuable insights into the dynamics of blocking and its predictability. In this paper, we employ the conditional nonlinear optimal perturbation (CNOP) method, which was initially proposed by Mu et al. (2003), to explore the most influential perturbations and their effects.

In this scenario, we define $B(t,x;\cdot,\cdot,\cdot)$ as the reference blocking amplitude with time evolution in the configuration space, where the three dots represent the three factors influencing the motion of blocking as mentioned previously. Given the initial blocking amplitude $B_0$, the synoptic-scale eddies $F_0$, and the background westerly wind $U$, we can express the blocking amplitude as $B(t,x;B_0,F_0,U)$. To quantify the magnitude of the blocking amplitude $B$, we utilize the standard mass norm, also known as the energy norm (Farrell and Ioannou, 1996). This norm is defined as

$$\|B(t)\| = \left( \int_{-L_x}^{L_x} |B(t,x)|^2 dx \right)^{\frac{1}{2}} \tag{11}$$

where $L_x$ represents the limits of integration. Similarly, for the synoptic-scale eddies, we also employ the standard mass norm given by

$$\|F\| = \left( \int_{-L_x}^{L_x} |F(x)|^2 dx \right)^{\frac{1}{2}}. \tag{12}$$

To enhance convenience in notation, we simplify the representation of the blocking amplitude as $B(t;B_0,F_0,U)$ by ignoring

the less commonly used variable $x$, which helps to avoid any confusion and improve the overall understanding and readability of the paper.

---

[2]In this study, the blocking's motion is primarily determined by the meridional gradient of potential vorticity ($PV_y$), which is influenced by the westerly wind $U$ and its meridional shear $U_{yy}$. However, for the specific focus of this paper, only the 1-dimensional forced NLS equation is considered, and therefore the meridional shear of the westerly wind $U_{yy}$ is disregarded. As a result, the main factor affecting the blocking's motion is the westerly wind $U$.



### 2.2.1 CNOP of initial blocking amplitude

If we consider the initial blocking amplitude as $B_0 + b_0$, where $b_0$ represents a perturbation of the initial blocking amplitude $B_0$, then the reference blocking amplitude at time $T$ can be expressed by $B(T; B_0 + b_0, F_0, U)$. Therefore, the two reference

blocking amplitudes are given by $B(T; B_0, F_0, U)$ and $B(T; B_0 + b_0; F_0, U)$. Based on the conditions of the synoptic-scale eddies $F_0$ and the westerly wind $U$, we can formulate the objective function for the initial perturbation $b_0$ about the initial blocking amplitude $B_0$ as

$$J(b_0; B_0, F_0, U) = \|B(T; B_0 + b_0, F_0, U) - B(T; B_0, F_0, U)\|^2 . \tag{13}$$

The CNOP can then be computed by maximizing $J(b_0; B_0, F_0, U)$ while ensuring that the constraint $\|b_0\| \leq \rho$ is satisfied,

where $\rho$ is a predetermined value that sets the upper limit for the norm of $b_0$. In other words, our goal is to find the optimal solution that maximizes $J(b0; B0, F0, U)$ while adhering to the constraint of $\|b_0\| \leq \rho$, expressed as

$$\max_{\|b_0\| \leq \rho} J(b_0; B_0, F_0, U). \tag{14}$$

By abbreviating $J(b_0; B_0, F_0, U)$ as $J(b_0)$, we can simplify and make it more convenient for subsequent discussions or calculations. This abbreviation allows us to refer to $J(b_0)$ more easily and conveniently without losing any generality.

### 225   2.2.2 CNOP of preexisting synoptic-scale eddies

In a similar manner, if we consider the synoptic-scale eddies as $F_0 + f_0$, where $f_0$ represents a perturbation of the preexisting synoptic-scale eddies $F_0$, then the reference blocking amplitude at time $T$ can be expressed by $B(T; B_0, F_0 + f_0, U)$. Consequently, we have two reference blocking amplitudes $B(T; B_0, F_0, U)$ and $B(T; B_0; F_0 + f_0, U)$. Based on the conditions of the initial blocking amplitude $B_0$ and the westerly wind $U$, we can formulate the objective function for the perturbation $f_0$ about

the background synoptic-scale eddies $F_0$ as

$$J(f_0; B_0, F_0, U) = \|B(T; B_0, F_0 + f_0, U) - B(T; B_0, F_0, U)\|^2 . \tag{15}$$

The CNOP can then be stated as maximizing $J(f_0; B_0, F_0, U)$ while ensuring that the constraint $\|f_0\| \leq \rho$ is satisfied, where $\rho$ is a predetermined value that sets the upper limit for the norm of $f_0$. In other words, we aim to find the optimal solution that maximizes $J(f0; B0, F0, U)$ while adhering to the constraint of $\|f_0\| \leq \rho$, expressed as

$$\max_{\|f_0\| \leq \rho} J(f_0; B_0, F_0, U). \tag{16}$$

It is also mentioned here that we shorten the notation $J(f_0; B_0, F_0, U)$ as $J(f_0)$ for convenience.

### 2.2.3 Variations in the westerly wind

By utilizing the CNOP approach, we can compute the optimal disturbance of the initial blocking amplitude, denoted as $b_0$ in eq. (14), and the optimal disturbance of the preexisting synoptic-scale eddies, denoted as $f_0$ in eq. (16). To explore the





influence of changes in the westerly wind $U$ on these optimal disturbances, we can assign different values to the westerly wind $U$ and observe the resulting changes, which allows us to understand how variations in the westerly wind affect the optimal disturbances.

### 2.2.4  Numerical implementation

In the theoretical analysis, the optimization problems concerning the optimal perturbations $b_0$ and $f_0$, as described in eq. (14)
and eq. (16), are directly derived from the forced NLS equation (9), which is considered as an infinite-dimensional model. However, when implementing it numerically on a computer, the optimization problems, (14) and (16), are reduced to finite-dimensional ones.

In this paper, we adopt the same method and numerical settings as described in (Luo, 2005; Luo et al., 2014, 2019) to study the time evolution of the blocking amplitude $B$. To numerically simulate the forced NLS equation with the periodic boundary
condition (9), we utilize the high-order split-step Fourier scheme developed in (Muslu and Erbay, 2005), which is known for its excellent performance. We set the nondimensional grid parameters, $\Delta x = 0.2296$ as the spatial grid size ($d = 101$) and $\Delta t = 0.0864$ as the time step. Additionally, the boundary parameter is set as $L_x = 11.48$ and the initial blocking amplitude is set as $B_0 = 0.4$. To compute the CNOP, we conventionally employ the second spectral projected gradient method (SPG2) proposed in (Birgin et al., 2000). The standard numerical gradient is computed with the step size $\epsilon = 10^{-8}$. The energy norm of the
blocking amplitude is numerically set as

$$\|b_0\| \approx \left( \sum_{i=1}^{d} b_{0,i}^2 \right)^{\frac{1}{2}} \sqrt{\Delta x} \le \rho = \gamma \sqrt{\Delta x}.$$

It is worth noting that an important observation from the empirical study (Breeden et al., 2020) is that the intensification of a blocking event often reaches its maximum around 10 days from the onset. Therefore, in this analysis, the prediction time is set at day $T = 10$ (unit: day). In line with the approach used in the previous studies (Luo, 2005; Luo et al., 2014, 2019), we set the
initial blocking amplitude as $B_0 = 0.4$ and the preexisting synoptic-scale eddies as $F_0 = a \exp \left[ -\mu \epsilon^2 (x + x_T)^2 \right]$.

### 3  Optimal disturbance of the initial blocking amplitude

In this section, the main objective is to investigate the optimal disturbance of the initial blocking amplitude. Our aim is to gain a better understanding of spatial patterns and nonlinear growth that are associated with this disturbance. Additionally, we also explore how the total blocking evolves as the optimal disturbance increases in size. Furthermore, we analyze the time-delay
effect of the optimal initial disturbance and its relation with predictability.

### 3.1  Spatial pattern and nonlinear growth

In the given text, our goal is to numerically compute the optimal disturbance of the initial blocking amplitude, denoted as $b_0$, which involves considering the governing forced NLS equation with the periodic boundary condition (2.9) and the initial blocking amplitude $B_0 = 0.4$. This computation can be achieved by maximizing the constrained objective function (13). Increasing



the size of the optimal disturbance allows for a more in-depth analysis of the numerical performance of spatial patterns. These spatial patterns are visualized in Figure 1, clearly representing how the spatial pattern varies with the incremental increase of the size parameter $\gamma$. It is truly fascinating to observe the spatial pattern of the optimal disturbance of the initial blocking amplitude in relation to the slow-varying preexisting synoptic-scale eddies $F_0 = a_0 \exp\left[-\mu\epsilon^2(x + x_T)^2\right]$. In Figure 1, we can clearly see a bulge around the zonal location $x = -x_T$, accompanied by two small dents beside it, resembling a solitary wave, as

noted in (Zabusky and Porter, 2010). As we increase the size parameter $\gamma$ incrementally from 0.25 to 1, the optimal disturbance exhibits a more pronounced solitary wave-like behavior. Specifically, the bulge becomes highly concentrated around the zonal center $x = -x_T$, with a sharply rising peak. This suggests that, in the context of blocking events in the real world, the largest deviation in the initial blocking amplitude $B_0$ occurs due to a positive incremental increase in the vicinity of the zonal location $x = -x_T$, which corresponds to the location of the synoptic-scale eddies acting as external forces. Additionally, the center of

the optimal disturbance of the initial blocking amplitude is slightly offset to the left of the zonal center $x = -x_T$, as depicted in Figure 1.

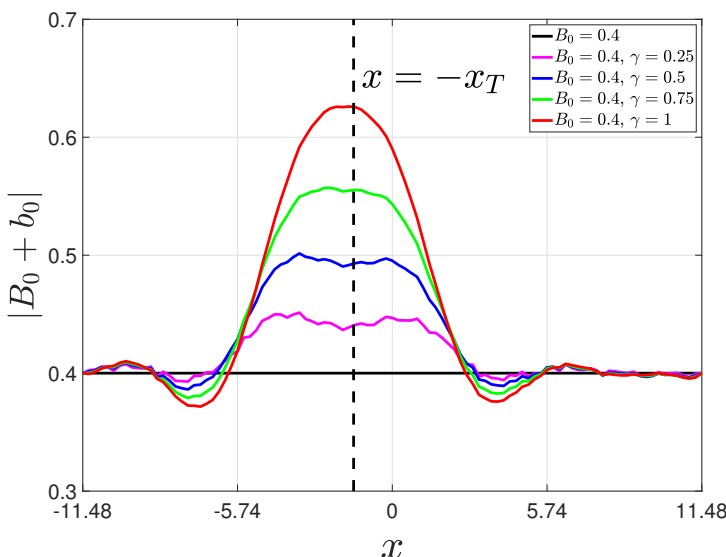

**Figure 1.** Spatial patterns (nondimensionalization) of the optimal disturbance $b_0$ under the initial blocking amplitude $B_0 = 0.4$ varies with the incremental increase of the size parameter $\gamma$.

Then, we utilize the energy norm (11) to characterize how the nonlinear growth of the optimal disturbance varies as the size increases, that is,

$$\frac{\|b(t)\|^2}{\Delta x} = \frac{\|B(t; B_0 + b_0, F_0, U) - B(t; B_0, F_0, U)\|^2}{\Delta x}, \tag{17}$$

where the nonlinear evolution of the optimal disturbance $\frac{\|b(t)\|^2}{\Delta x}$ measures the difference between the blocking amplitudes $B$ at time $t$, taking into account both the initial blocking amplitude $B_0$ and the most perturbed initial blocking amplitude $B_0 + b_0$,





while keeping the synoptic-scale eddies $F_0$ and the westerly wind speed $U$ fixed. This allows for a comparison of the effects of the optimal disturbance on the blocking amplitude. The numerical performance of the nonlinear growth behavior is visualized in Figure 2a. To further understand the nonlinear growth behavior, it is indeed important to investigate the relative nonlinear

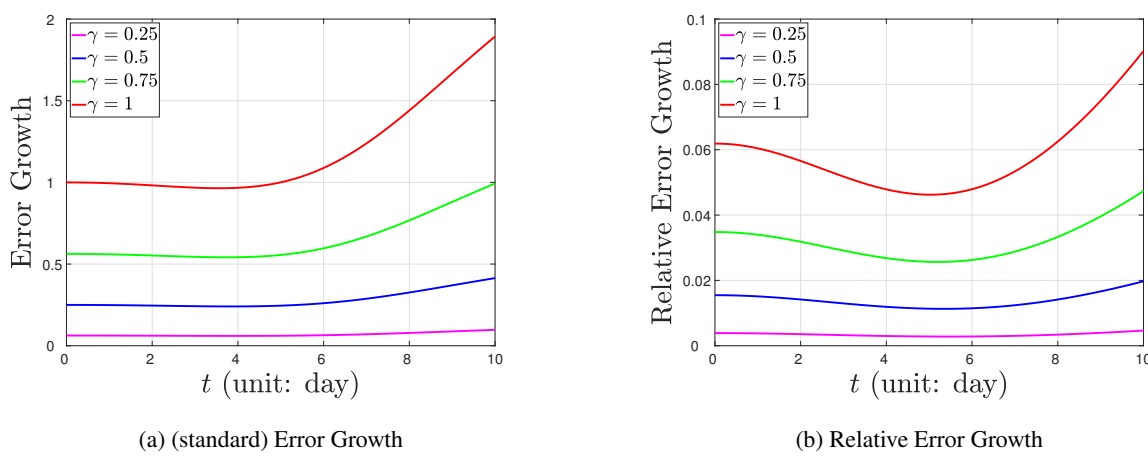

(a) (standard) Error Growth                                    (b) Relative Error Growth

**Figure 2.** Nonlinear growth of the optimal disturbance given by (17) and (18) varies with the incremental increase of the size parameter $\gamma$.

growth of the optimal disturbance. This can be done by comparing the nonlinear growth of the optimal disturbance $\frac{\|b(t)\|^2}{\Delta x}$ with the blocking amplitude $\frac{\|B(t)\|^2}{\Delta x}$ using their ratio, that is,

$$\frac{\|b(t)\|^2}{\|B(t)\|^2} = \frac{\|B(t;B_0+b_0,F_0,U) - B(t;B_0,F_0,U)\|^2}{\|B(t;B_0,F_0,U)\|^2}. \tag{18}$$

By examining the relative nonlinear growth of the optimal disturbance (18), it is observed that the nonlinear growth of the optimal disturbance is slower compared to the growth of the blocking amplitude during the period of blocking growth, while

faster during other periods. The numerical performance is shown in Figure 2b. From both the subfigures in Figure 2, it is evident that there is a fixed turning-time point in the nonlinear growth of the optimal disturbance. Prior to reaching this turning-time point, the nonlinear growth is relatively slow. However, once the turning-time point is reached, the nonlinear growth accelerates rapidly, and its growth pattern undergoes a significant change. It is also noteworthy that as the size parameter $\gamma$ increases incrementally, the turning-time point occurs earlier and the nonlinear growth of the optimal disturbance becomes

more pronounced. This suggests that the size of the optimal disturbance has a positive impact on the timing and magnitude of its nonlinear growth. Meanwhile, we show the ratios of the nonlinear growth of the optimal disturbance in terms of norm squares in Table 2, where it is observed that the ratio increases as the size increases incrementally, further indicating a rising growth rate. The quantitative evidence in Table 2 supports the idea that the size of the optimal disturbance has a positive effect on accelerating its nonlinear growth. Additionally, we compare the ratio of the relative nonlinear growth to further quantify

the positive effect of the size of the optimal disturbance on its nonlinear growth, which is demonstrated in Table 3. It is worth noting that the nonlinear growth behavior observed in Table 3 aligns with the findings in Table 2. This consistency provides





|  | $\gamma = 0.25$ | $\gamma = 0.5$ | $\gamma = 0.75$ | $\gamma = 1$ |
|---|---|---|---|---|
| $\frac{\|b(10)\|^2}{\|b(0)\|^2}$ | 1.5616 | 1.6568 | 1.7664 | 1.8935 |

**Table 2.** The ratio of the nonlinear growth of the optimal disturbance in terms of norm squares, $\|b(10)\|^2/\|b(0)\|^2$.

|  | $\gamma = 0.25$ | $\gamma = 0.5$ | $\gamma = 0.75$ | $\gamma = 1$ |
|---|---|---|---|---|
| $\frac{\|b(0)\|^2}{\|B(0)\|^2}$ | 0.39% | 1.55% | 3.48% | 6.19% |
| $\frac{\|b(10)\|^2}{\|B(10)\|^2}$ | 0.47% | 1.97% | 4.73% | 9.02% |
| $\frac{\|b(10)\|^2}{\|B(10)\|^2} / \frac{\|b(0)\|^2}{\|B(0)\|^2}$ | 1.2051 | 1.2710 | 1.3592 | 1.4572 |

**Table 3.** The relative nonlinear growth of the optimal disturbance in terms of norm squares, $\|b(0)\|^2/\|B(0)\|^2$ and $\|b(10)\|^2/\|B(10)\|^2$, and the ratio between them.

further evidence to support the idea that the size of the optimal disturbance does have a positive effect on accelerating its nonlinear growth.

By normalizing the initial conditions for the growth curves in Figure 2, we demonstrate that increasing the size of the optimal disturbance can indeed accelerate its nonlinear growth. The normalization for the nonlinear growth is given by $\frac{\|b(t)\|^2}{\|b(0)\|^2}$, while the relative nonlinear growth is given by $\frac{\|b(t)\|^2}{\|B(t)\|^2} / \frac{\|b(0)\|^2}{\|B(0)\|^2}$. This normalization indeed provides a clearer visualization of the nonlinear growth patterns of the optimal disturbances in terms of norm square, as shown in Figure 3. This normalization allows us to observe how fast the optimal disturbance grows over time as the size parameter $\gamma$ increases incrementally from 0.25 to 1. Furthermore, Figure 3 shows that the nonlinear evolution of the optimal disturbances, transitioning from units to ratios, which

aligns with the data shown in Table 2 and Table 3. This comprehensive representation provides us with a better understanding of how the optimal disturbances change and grow throughout the process. By comparing Figure 3a and Figure 3b, it further supports the idea that the growth of the optimal disturbance is weaker than that of the blocking amplitude during the period of blocking growth, while faster during other periods.

### 3.2 Temporal evolution of blocking under the optimal disturbance

After analyzing the spatial patterns and the nonlinear growth of the optimal disturbance, it would indeed be valuable to explore how the motion of the blocking is influenced by the optimal disturbance. Understanding the dynamic relationship between the optimal disturbance and the evolution of the blocking can provide further insights into the impact of the optimal disturbance on the overall behavior of the blocking system. In particular, it would be interesting to investigate how the blocking evolves with time when the optimal disturbance is added to the initial blocking amplitude.

Based on the provided expressions (7a) and (7b), the blocking wavy anomaly is represented as $\psi_B = \epsilon\psi_1$ and the associated zonal-mean anomaly is represented as $\psi_m = \epsilon^2\psi_2$. Additionally, based on eq. (7c), the streamfunction of the preexisting





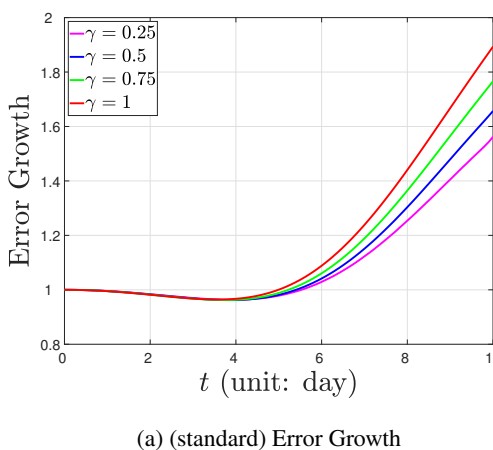
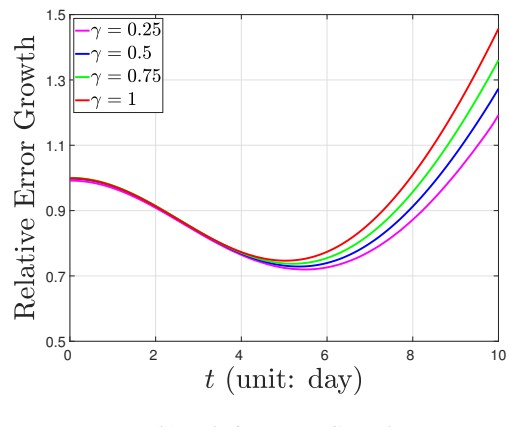

(a) (standard) Error Growth  (b) Relative Error Growth

**Figure 3.** Nonlinear growth of the optimal disturbance given by (17) and (18) under initial normalization varies with the increase of the size parameter $\gamma$.

synoptic-scale eddies is approximated by $\psi' \approx \epsilon^{\frac{3}{2}}\psi'_1 + \epsilon^{\frac{5}{2}}\psi'_2$.[3] According to the expression (2), the total streamfunction can be approximated as $\psi_T = \overline{\psi} + \psi + \psi' \approx \overline{\psi} + \psi_B + \psi_m + \epsilon^{\frac{3}{2}}\psi'_1 + \epsilon^{\frac{5}{2}}\psi'_2$. In Section 2.1, it is derived that the blocking amplitude is governed by the forced NLS equation with the periodic boundary condition (9). As mentioned in (Luo et al., 2019), the evolution of the instantaneous total streamfunction $\psi_T$ is entirely dependent on the blocking amplitude throughout the whole life of the blocking. In Figure 4, we show the evolution of the instantaneous total streamfunction $\psi_T$ when the initial blocking amplitude is added by the optimal disturbances. This visualization provides a clear representation of how the total streamfunction changes as the size parameter $\gamma$ increases incrementally. Figure 4a visualizes the evolution of the instantaneous total streamfunction without any perturbations, highlighting two predominant phenomena commonly associated with the blocking events: eddy straining and wave breaking. These phenomena, as described in (Shutts, 1983; Pelly and Hoskins, 2003), are known to play a significant role in the occurrence of blocking events. By comparing Figure 4a to Figure 4b and Figure 4c from left to right, it is observed that as the size parameter $\gamma$ increases, the phenomena of eddy straining and wave breaking become more prominent. However, both the position and the period of the blocking remain almost unchanged. During the maintenance period of the blocking, the intensification of eddy straining and wave breaking becomes extremely dominant, but their positions and periods remain mostly invariant. From a physics perspective, it appears that the optimal disturbance of initial blocking amplitude tends to intensify the strength of the blocking without causing any other significant changes. This intensification becomes more pronounced as the size of the optimal disturbance increases. This observation provides valuable insights into the motion of the blocking and the role of the size of the optimal disturbance in their intensification. It is possible that the nonlinear overgrowth caused by the optimal disturbance of the initial blocking amplitude, as mentioned in (Bengtsson, 1981; Tibaldi and Molteni, 1990; Burroughs, 1997), could be a contributing factor to the frequent occurrence of extreme weather events and the subse-

---

[3]The derivation of the deformed synoptic-scale eddy $\psi'_2$ is so tedious and circumstantial that we postpone it to the supplementary materials.







(a) $B_0 = 0.4$          (b) $B_0 = 0.4$, $\gamma = 0.5$          (c) $B_0 = 0.4$, $\gamma = 1$

**Figure 4.** Nonlinear evolution of the instantaneous total streamfunction field $\psi_T$ when the initial blocking amplitude is added by the optimal disturbance with the incremental increase of the size parameter $\gamma$.

quent decrease in predictability. This suggests that the nonlinear growth of the optimal disturbance, dominated by its size, may lead to the complex dynamics of the blocking.



### 3.3 Less predictability on the medium-range

Weather prediction systems have indeed made significant progress over the years, thanks to advancements in technology, data
collection, and modeling techniques. These advancements have greatly improved the accuracy and reliability of weather fore-
casts. However, despite these improvements, accurately predicting blocking on the medium-range weather timescale remains a
challenge, as highlighted in (Kautz et al., 2022). Additionally, Hamill and Kiladis (2014) have found that forecast errors tend to
be larger for European blocking compared to other regimes, particularly during the transition phases into or out of a blocking
regime. These findings emphasize the complexity and difficulty in accurately forecasting blocking events.
It sounds like a reasonable approach for conducting a numerical experiment using the NMI model to explore whether there
are larger forecast errors caused by the optimal disturbance. To start, we run the forced NLS equation (9) for 5 days without any
perturbations. After that, using the blocking amplitude on the 5th day as the initial condition, our goal is to obtain the optimal
disturbance during the period from 5 to 15 days. The spatial pattern of the optimal disturbance at a later stage, compared with
the initial stage, is shown in Figure 5a, where it is observed that the peak of the solitary wave slightly offsets to the right and
becomes sharper. The nonlinear growth and relative nonlinear growth of the optimal disturbances are depicted in Figure 5b
and Figure 5c, as described by eq. (17) and eq. (18), respectively. It is worth noting that when the time interval $T$ and the size
parameter $\gamma = 1$ are fixed, the optimal disturbance at a later stage leads to a larger error, which is demonstrated quantitatively
by the ratios between the later stage and the initial stage are 1.8776 for the nonlinear growth and 1.8781 for the relative
nonlinear growth, respectively. Additionally, the experiment reveals that the nonlinear evolution of the optimal disturbance is
more pronounced during the decay of the blocking, while the error is smaller during the maintenance of the blocking. This
finding aligns with the less predictability of blockings on the medium-range, as mentioned in (Hamill and Kiladis, 2014;
Ferranti et al., 2015; Zhang et al., 2019). This suggests that the presence of the optimal disturbance at a later stage, as observed
in the numerical experiment, can contribute to larger forecast errors in predicting blockings.

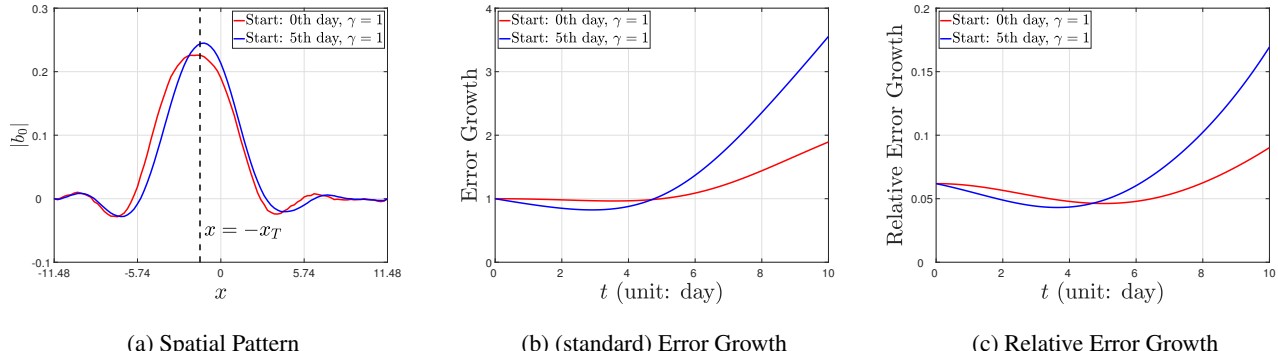

(a) Spatial Pattern  (b) (standard) Error Growth  (c) Relative Error Growth

**Figure 5.** Spatial pattern and nonlinear growth of the optimal disturbance at a later stage in comparison to the baseline at the initial stage.





## 4 Optimal disturbance of the preexisting synoptic-scale eddies

In this section, our focus is on investigating the optimal disturbance of the preexisting synoptic-scale eddies. Our aim is to understand the spatial patterns and the nonlinear growth of the error associated with this disturbance. We also explore how the total blocking evolves as the optimal disturbance incrementally increases in size. Furthermore, we analyze the time-delay effect of the optimal disturbance and its relation with predictability.

### 4.1 Spatial pattern and nonlinear growth

In this given context, our goal is to numerically compute the optimal disturbance of the preexisting synoptic-scale eddies, denoted as $f_0$. This computation is based on the forced NLS equation with the periodic boundary condition (9) and the preexisting synoptic-scale eddies $F_0 = a_0 \exp\left[-\mu\epsilon^2(x+x_T)^2\right]$. To achieve this, we can maximize the constrained objective function (15). Increasing the size of the optimal disturbance, denoted by the parameter $\gamma$, allows for a more in-depth analysis of the numerical performance of spatial patterns. In Figure 6, we provide a clear visualization of the spatial patterns, showcasing how they vary

as the size parameter $\gamma$ increases incrementally. The optimal disturbance is observed to concentrate sharply around a slight offset to the left of the zonal center $x = -x_T$, which appears as a sharp bulge with two small dents on either side and two small bulges beside them. As the size parameter $\gamma$ increases incrementally from 0.25 to 1, the optimal disturbance becomes even more pronouncedly sharp. Specifically, the bulge becomes highly concentrated, resembling a sharply rising peak, which appears to be more predominant than the optimal disturbance of the initial blocking amplitude. This suggests that, in the context

of blocking events in the real world, the largest deviation in the preexisting synoptic-scale eddies concentrates sharply around a slight offset to the left of the zonal center $x = -x_T$.

Next, we explore the nonlinear growth of the error caused by the optimal disturbance of the preexisting synoptic-scale eddies. The energy norm (11) is utilized to quantify this growth as

$$\frac{\|b(t)\|^2}{\Delta x} = \frac{\|B(t; B_0, F_0 + f_0, U) - B(t; B_0, F_0, U)\|^2}{\Delta x}, \tag{19}$$

which measures the difference between the blocking amplitudes $B$ at time $t$, considering both the preexisting synoptic-scale eddies $F_0$ and the most perturbed preexisting synoptic-scale eddies $F_0 + f_0$, while keeping the initial blocking amplitude $B_0$ and the westerly wind speed $U$ fixed. This allows for a comparison of the effects of the optimal disturbance on the blocking amplitude. The numerical performance of the nonlinear growth behavior is visualized in Figure 7a. It is intriguing to see that the nonlinear growth of the error caused by optimal disturbance of the preexisting synoptic-scale eddies exhibits a striking

rise as the size parameter $\gamma$ increases incrementally. This behavior seems to be fully distinct from the nonlinear growth of the optimal disturbance of the initial blocking amplitude, as shown in Figure 2a and Figure 3a. The phenomenon of the error growing several times aligns with weather predictions in the real world, as mentioned by Zhang et al. (2019). Similarly, we also explore the relative nonlinear growth of the error generated by the optimal disturbance, which can be calculated by taking the ratio between the nonlinear growth of the error $\frac{\|b(t)\|^2}{\Delta x}$ and the blocking amplitude $\frac{\|B(t)\|^2}{\Delta x}$ as

$$\frac{\|b(t)\|^2}{\|B(t)\|^2} = \frac{\|B(t; B_0, F_0 + f_0, U) - B(t; B_0, F_0, U)\|^2}{\|B(t; B_0, F_0, U)\|^2}. \tag{20}$$





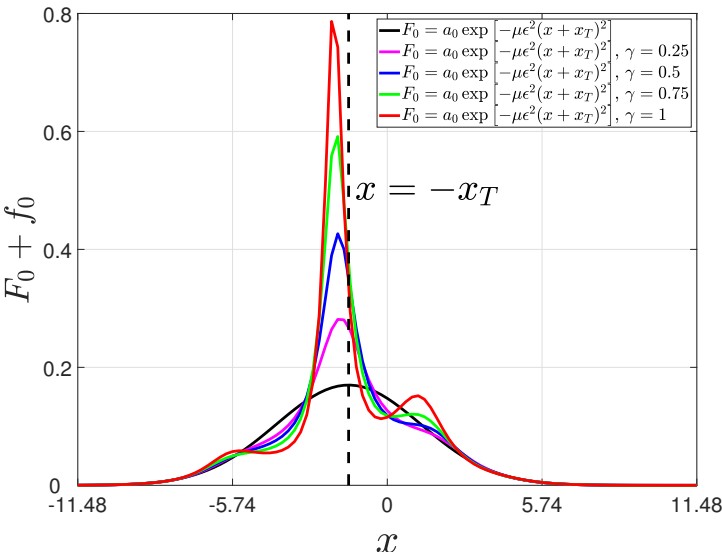

**Figure 6.** Spatial patterns (nondimensionalization) of the optimal disturbance $f_0$ under the preexisting synoptic-scale eddies varies with the increase of the size parameter $\gamma$.

It is interesting to note that in Figure 7b, the relative nonlinear evolution of the error caused by the optimal disturbance also exhibits a significant growth, which is consistent with the nonlinear growth observed in Figure 7a. Comparing these subfigures in Figure 7 with the nonlinear growth of the optimal disturbance of the initial blocking amplitude shown in Figure 2 and Figure 3 can help identify their differences. It appears that the optimal disturbance of the preexisting synoptic-scale eddies results in a

considerably large error growth. This finding provides further insight into the role played by the fast-moving short-lived (high-frequency) synoptic-scale eddies in the blocking system, which has been previously studied in (Berggren et al., 1949; Shutts, 1983; Hoskins et al., 1985; Luo et al., 2014, 2019). This highlights the potential impact of such disturbance on blockings, which could be a probable cause of weather extremes and reduce predictability. To further provide a more comprehensive characterization of the striking growth of the error caused by the optimal disturbance of the preexisting synoptic-scale eddies, it

is indeed important to include the quantitative measurements of the nonlinear growth and the relative nonlinear growth. These quantitative measurements are calculated and presented in Table 4, which further provides the evidence that the nonlinear growth and the relative nonlinear growth appear more pronouncedly sharp as the size parameter $\gamma$ increases incrementally from 0.25 to 1 .

## 4.2   Temporal evolution of blocking under the optimal disturbance

It is indeed a valuable step to explore how the motion of the blocking is influenced by the optimal disturbance. Understanding the dynamic relationship between the optimal disturbance and the evolution of the blocking can provide further insights into





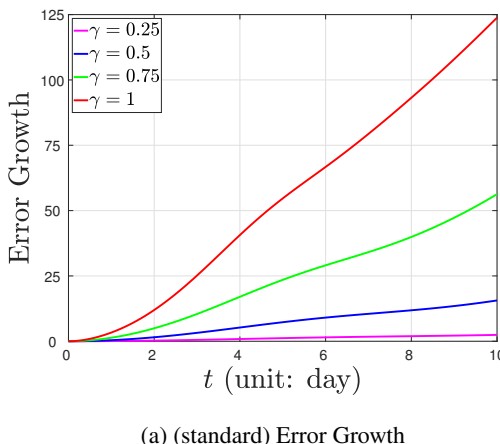
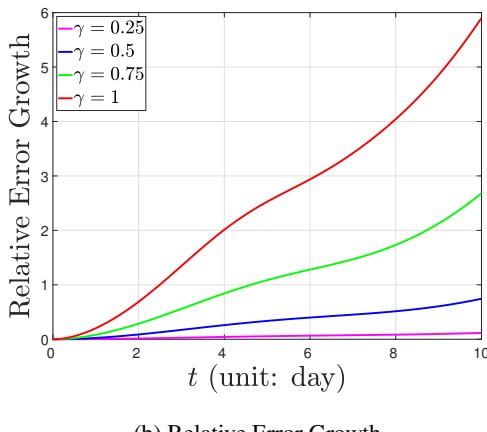

(a) (standard) Error Growth                                  (b) Relative Error Growth

**Figure 7.** Nonlinear growth of the error generated by the optimal disturbance of the preexisting synoptic-scale eddies given by (19) and (20) varies with the increase of the size parameter $\gamma$.

|  | $\gamma = 0.25$ | $\gamma = 0.5$ | $\gamma = 0.75$ | $\gamma = 1$ |
|---|---|---|---|---|
| $\frac{\|b(10)\|^2}{\Delta x}$ | 2.4371 | 15.6206 | 56.2985 | 123.8228 |
| $\frac{\|b(10)\|^2}{\|B(10)\|^2}$ | 0.1161 | 0.7443 | 2.6826 | 5.9000 |

**Table 4.** The nonlinear growth and relative nonlinear growth of the error caused by the optimal disturbance of the preexisting synoptic-scale eddies in terms of norm squares.

the impact of the optimal disturbance on the overall behavior of the blocking system. In particular, it would be interesting to investigate how the blocking evolves with time when the optimal disturbance is added to the preexisting synoptic-scale eddies.

In Figure 8, we depict the evolution of the instantaneous total streamfunction $\psi_T$ when the preexisting synoptic-scale ed-
dies are added by the optimal disturbances. This visualization offers a clear representation of how the total streamfunction changes as the size parameter $\gamma$ increases incrementally. Figure 8a is the same as Figure 4a, which illustrates that the evolution of the instantaneous total streamfunction without any perturbations. Taking a corresponding comparison, Figure 8a with Figure 4a, Figure 8b with Figure 4b, Figure 8c with Figure 4c, it becomes evident that as the size parameter $\gamma$ increases incrementally, the phenomena related to blocking, eddy straining and wave breaking, become more prominent in comparison
to the motion of the blocking added by the optimal disturbance of the initial blocking amplitude. Additionally, both the position and the period of the blocking exhibit significant changes and become chaotic. This observation indicates that the behavior of the blocking indeed becomes more unpredictable and less stable when perturbations occur in the preexisting synoptic-scale eddies. In other words, this highlights the sensitivity of the blocking to perturbations of the preexisting synoptic-scale eddies, which can potentially lead to weather extremes and pose challenges in accurately predicting them, as mentioned in (Bengtsson,
1981; Tibaldi and Molteni, 1990; Burroughs, 1997). This further suggests that the error caused by the optimal disturbance







**Figure 8.** Nonlinear evolution of the instantaneous total streamfunction field $\psi_T$ when the preexisting synoptic-scale eddies are added by the optimal disturbance with the incremental increase of the size parameter $\gamma$.

of the preexisting synoptic-scale eddies can indeed contribute to the complex dynamics of the blocking. As the size of the disturbance increases, the complexity of the blocking can also increase.



### 4.3 Less predictability on the medium-range

Similarly, it is also a valuable endeavor to conduct the numerical experiment to explore the potential impact of the optimal
disturbance of preexisting synoptic-scale eddies on forecast errors. Specifically, we investigate whether there are larger forecast
errors at a later stage by taking a comparison of optimal disturbances between different time ranges, such as an early stage
from 0 to 10 days and a later stage from 5 to 15 days. The spatial pattern of the optimal disturbance at a later stage, compared
with an early stage, is shown in Figure 9a, where it is observed that the concentration distribution as a sharp peak slightly
offsets to the right and flattens. Additionally, the nonlinear growth and relative nonlinear growth of the error caused by the

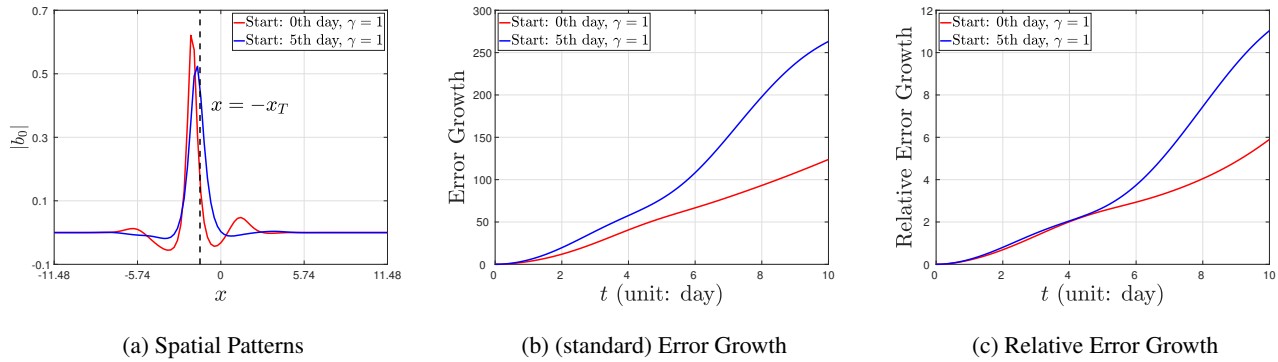

| (a) Spatial Patterns | (b) (standard) Error Growth | (c) Relative Error Growth |

**Figure 9.** Spatial patterns and nonlinear growth of the error caused by the optimal disturbance at a later stage in comparison to the baseline
at the initial stage.

optimal disturbances are depicted in Figure 9b and Figure 9c, as described by eq. (19) and eq. (20), respectively. Compared
with the growth patterns of the optimal disturbance of the initial blocking amplitude shown in Figure 5b and Figure 5c, it
appears that the error caused by the optimal disturbance of the preexisting synoptic-scale eddies grows more predominantly
during the decay period of the blocking. It is worth noting that when the time interval $T$ and the size parameter $\gamma = 1$ are fixed,
the optimal disturbance at a later stage leads to a larger error. This difference in error is further quantitatively highlighted by
the ratios between the later stage and the initial stage, 2.1256 for the nonlinear growth and 1.8706 for the relative nonlinear
growth. Additionally, the experiment reveals that the nonlinear evolution of the error caused by the optimal disturbance of
the preexisting synoptic-scale eddies exhibits a sharp growth during the decay of the blocking. This finding about the optimal
disturbance of the synoptic-scale eddies aligns with that of the initial blocking amplitude shown in Section 3.3, which leads
to the less predictability of blockings on the medium-range, as mentioned in (Hamill and Kiladis, 2014; Ferranti et al., 2015;
Zhang et al., 2019). This also support the idea that the presence of the optimal disturbance at a later stage, as observed in the
numerical experiment, can contribute to larger forecast errors in predicting blockings.





# 5 The impact of the background westerly wind

In this section, we take several numerical experiments to explore how the background westerly wind affects the optimal
disturbances of both the initial blocking amplitude and the preexisting synoptic-scale eddies. Specifically, we investigate their
spatial patterns and nonlinear growth of the error caused by the optimal disturbances, which provide insights into the influence
of the background westerly wind on these weather phenomena.

### 5.0.1 The optimal disturbance of the initial blocking amplitude

In our numerical experiment, we fix the size parameter of the optimal disturbance as $\gamma = 1$. Decreasing the westerly wind speed
$U$ from 1.1 to 0.3 with a decrement of 0.2 allows us to explore how the variation of wind speed affects the optimal disturbance
of the initial blocking amplitude. The spatial patterns, as shown in Figure 10a, indicate that the solitary wave-like pattern
becomes more concentrated, sharper, and gradually shifts to the right as the wind speed gradually dwindles. Additionally, the
growth patterns depicted in Figure 10b and Figure 10c reveal that the nonlinear growth and relative nonlinear growth become
gradually larger as the wind speed gradually dwindles. These observations suggest that the westerly wind speed $U$ plays a role

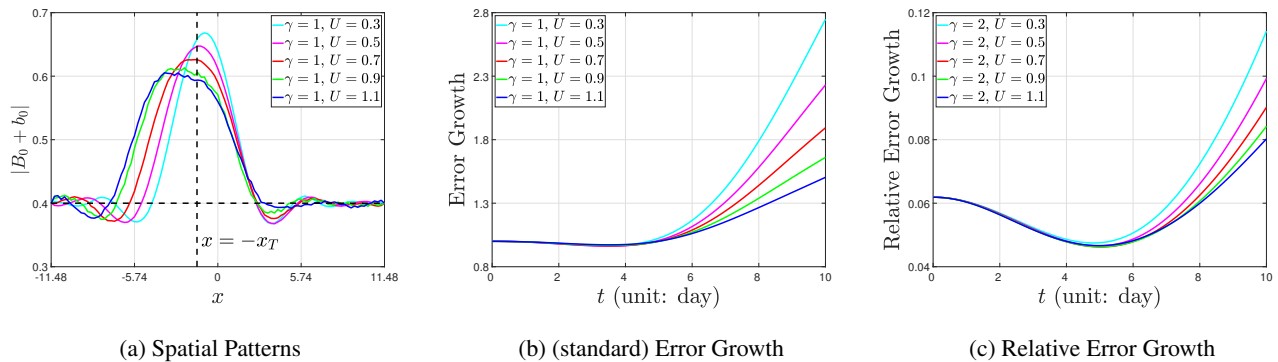

| (a) Spatial Patterns | (b) (standard) Error Growth | (c) Relative Error Growth |

**Figure 10.** Spatial patterns and nonlinear growth of the optimal disturbance of the initial blocking amplitude varies with the changes of the
background westerly winds.

shaping the spatial and growth patterns. Furthermore, we quantitatively show the nonlinear growth (17) and relative nonlinear
growth (18) at the prediction time $T = 10$ in Table 5. Indeed, it verifies that the nonlinear growth and relative nonlinear growth
become gradually larger as the wind speed gradually dwindles.

|  | $U = 0.3$ | $U = 0.5$ | $U = 0.7$ | $U = 0.9$ | $U = 1.1$ |
|---|---|---|---|---|---|
| $\dfrac{\|b(10)\|^2}{\Delta x}$ | 2.7481 | 2.2307 | 1.8935 | 1.6610 | 1.5036 |
| $\dfrac{\|b(10)\|^2}{\|B(10)\|^2}$ | 0.1142 | 0.0993 | 0.0902 | 0.0842 | 0.0802 |

**Table 5.** The growth patterns of the optimal disturbance of the initial blocking amplitude $B_0 = 0.4$.



The influence of the westerly wind on the nonlinear growth behavior, as depicted in Figure 10 and Table 5, aligns with the $PV_y$ theory for the NMI model proposed in (Luo et al., 2019). The relationship between the potential vorticity and the westerly wind, as described in (Pedlosky, 1987), can be expressed as $PV = f_0 + \beta y - U_y - F\overline{\psi}$. When considering a uniform westerly

wind, the meridional gradient of potential vorticity has a linear relation with the westerly wind, given by $PV_y = \beta + FU$. Based on the forced NLS equation (9) and the conditions of coefficients (10), we can deduce that the coefficient of the dispersive term is proportional to $PV_y$, i.e., $\lambda \propto PV_y = \beta + FU$, and the coefficient of the nonlinear term is inversely proportional to $PV_y$, i.e., $\delta \propto 1/PV_y = 1/(\beta + FU)$. Therefore, when the westerly wind weakens, the meridional gradient of potential vorticity decreases, resulting in the suppression of the dispersive effect and the intensification of the nonlinear effect. This results in an increase in

the nonlinear growth. Conversely, when the westerly wind strengthens, the meridional gradient of potential vorticity increases, resulting in the intensification of the dispersive effect and the suppression of the nonlinear effect, leading to a decrease in the nonlinear growth. Furthermore, as the coefficient of the nonlinear term is inversely proportional to $PV_y$, when the westerly wind speed gradually dwindles, the rate of increase in $PV_y$ becomes fast, causing the nonlinear growth rate to become large.

### 5.0.2 The optimal disturbance of the preexisting synoptic-scale eddies

Similarly, for the numerical experiment related to the preexisting synoptic-scale eddies, the size parameter of the optimal disturbance is also set as $\gamma = 1$. Then, we explore how the variation of the wind speed affects the optimal disturbance of the preexisting synoptic-scale eddies by decreasing the westerly wind speed $U$ from 1.1 to 0.3 with a decrement of 0.2. The spatial patterns, as shown in Figure 11a, demonstrate how the spatial patterns change as the wind speed varies. It is intriguing

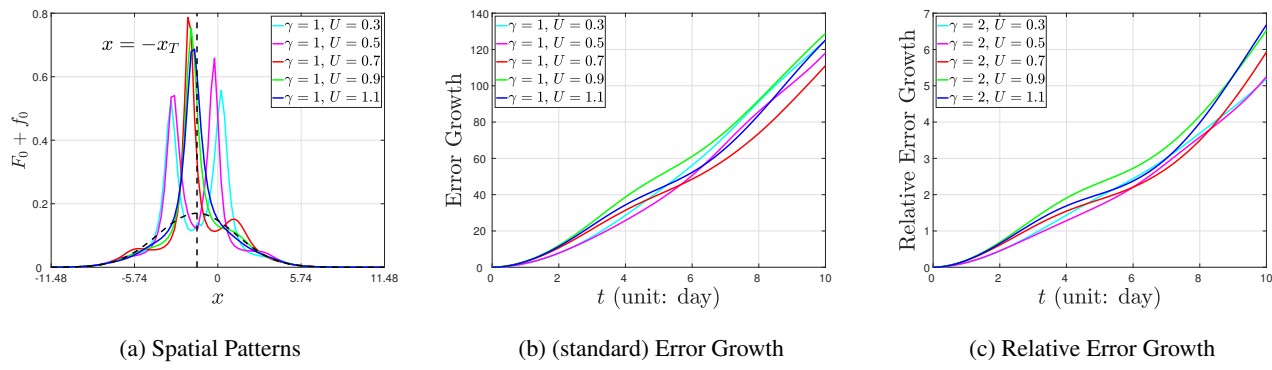

(a) Spatial Patterns        (b) (standard) Error Growth        (c) Relative Error Growth

**Figure 11.** Spatial patterns and nonlinear growth of the error caused by the optimal disturbance of the preexisting synoptic-scale eddies vary with the changes of the background westerly winds.

to observe the different behaviors of the peak-like pattern based on wind speed variations. When the wind speed is smaller

than the standard speed $U = 0.7$, the sharp peak-like pattern separates into two lower peaks on both sides. As the wind speed decreases further, the two peak-like pattern moves outside and becomes lower. On the other hand, when the wind speed is larger than the standard speed $U = 0.7$, the sharp peak-like pattern descends and shifts slightly to the right. As the wind speed increases further, the sharp peak-like pattern continues to descend and shift slightly to the right. However, it is worth noting





that the growth behaviors of the error caused by the optimal disturbance of the preexisting synoptic-scale eddies, regardless
of the nonlinear growth or the relative nonlinear growth, are rarely influenced by the variation of the wind speed, as shown
in Figure 11b and Figure 11c.

## 6  Summary and Discussion

Taking the barotropic NMI model developed in (Luo, 2000, 2005; Luo et al., 2014, 2019) as a basis, we have specifically
focused on exploring optimal disturbances of blocking by utilizing the CNOP approach in this paper. In the NMI model,
the motion of blocking is governed by the forced NLS equation (9), which provides a framework for studying the optimal
disturbances of the initial blocking amplitude and preexisting synoptic-scale eddies. Our analysis of the optimal disturbances
includes examining their spatial patterns, nonlinear growth patterns of the error caused by them, their influence on the motion
of the total blocking, and their time-delay effect. It is observed that the optimal disturbance of the initial blocking amplitude
has a well-behaved impact solely on the blocking amplitude without any other influence. Increasing the size of the optimal
disturbance indeed accelerates the nonlinear growth of the error. However, a striking phenomenon is observed in the optimal
disturbance of the preexisting synoptic-scale eddies, which leads to a significant increase in error growth. As the size of
the disturbance increases, the phenomena related to blocking, eddy straining and wave breaking, become highly noticeable.
Specifically, this results in significant changes in both the position and period of the blocking, leading to chaotic behavior.
This finding manifests that the blocking is extremely sensitive to perturbations of the fast-moving short-lived (high-frequency)
synoptic-scale eddies in the blocking system, as mentioned in (Bengtsson, 1981; Tibaldi and Molteni, 1990; Burroughs, 1997).
Furthermore, it highlights the role played by the fast-moving short-lived (high-frequency) synoptic-scale eddies in the blocking
system, as mentioned in (Berggren et al., 1949; Shutts, 1983; Hoskins et al., 1985; Luo et al., 2014, 2019). The perturbations
of these eddies may be a probable cause of weather extremes and can reduce predictability. Additionally, both the optimal
disturbances occurring at a later stage, particularly during the decay period of blocking, also contribute to accelerating the
nonlinear growth of the error. This has implications for the predictability of blockings on the medium-range, which aligns with
the practical weather prediction, as mentioned in (Hamill and Kiladis, 2014; Ferranti et al., 2015; Zhang et al., 2019). Finally,
we have analyzed the influences of the westerly wind on the optimal disturbances. Regarding the initial blocking amplitude,
the nonlinear evolution behavior indicates that the influence of the westerly wind on the optimal disturbance aligns with the
$PV_y$ theory proposed in (Luo et al., 2019). However, when considering the preexisting synoptic-scale eddies, it is observed that
the westerly wind has no impact on them.

In this paper, our main focus is on studying the optimal disturbances of the initial blocking amplitude and preexisting
synoptic-scale eddies, as well as the influence of the westerly wind. This study utilizes the 1-dimensional forced NLS equa-
tion that specifically considers the zonal direction. However, it is acknowledged that the meridional shear of the westerly
wind, represented as $PV_y = \beta + FU - U_{yy}$, also plays a significant role in the meridional gradient of potential vorticity. Previ-
ous studies, such as (Thorncroft et al., 1993), have observed that the meridional shear of the background westerly wind can
break up synoptic-scale anticyclones or cyclones. The $PV_y$ theory proposed in (Luo et al., 2019), it further suggests that it



affects the dispersive and nonlinear effects. When there is positive shear $U_{yy} > 0$, the dispersive effect is suppressed and the nonlinear effect is intensified. On the other hand, there is the negative shear $U_{yy} < 0$, the dispersive effect is intensified, and the nonlinear effect is suppressed. Therefore, it would be valuable to conduct further research to explore the optimal distur-
bances in the 2-dimensional NMI model. Additionally, the 3-dimensional baroclinic NMI model, developed in (Luo and Zhang, 2020a, b, 2021), considers the non-homogeneous vertical structure. Hence, it would indeed be intriguing to further explore how the optimal disturbances change and their influence on the growth of errors by taking into account the effect of the horizontal temperature gradients.

There are indeed other theories and models related to blocking, such as the local wave activity proposed in (Huang and
Nakamura, 2016), the traffic jam theory in (Nakamura and Huang, 2018), and the amplified Rossby wave theory in (Kornhuber et al., 2020). It would be intriguing to investigate how different types of perturbations in these models influence the growth of errors. Exploring the impact of these perturbations can provide valuable insights into the behavior and dynamics of blocking systems. Furthermore, Mu and Jiang (2008) started utilizing the CNOP approach in the T21L3 quasi-geostrophic model (Vannitsem and Nicolis, 1997) to investigate the initial perturbations that trigger blocking onset. It would be worth exploring the
influence of the perturbations by employing the CNOP approach in the real numerical weather prediction model, as suggested in (Zhang et al., 2019). Additionally, the planetary solitary waves are also large-scale important phenomena occurring in the atmosphere and ocean with diameters from a hundred kilometers to scale larger than the earth, such as vortices embedded in a shear zone, Rossby solitons, and equatorial Kelvin solitary waves among others (Rizzoli, 1982; Boyd, 2007). It would also be valuable to investigate the influence of the perturbations on their motion by employing the CNOP approach.

In the conclusion of this paper, we briefly discuss the perturbations of blocking in response to climate change, which is currently a hot topic, as suggested in (Woollings et al., 2018; Kautz et al., 2022). It is worth noting that numerical climate models have always faced challenges in accurately representing blocking events, since these models tend to underestimate both the occurrence and persistence of blocking events, as suggested in (Tibaldi and Molteni, 1990; d'Andrea et al., 1998; Davini and D'Andrea, 2016). It has also been observed that apparent improvements of blocking representation in a numerical
model can sometimes occur through compensation of errors, as mentioned in (Davini et al., 2017). Additionally, increasing the horizontal resolution of a numerical model can enhance the transient eddy forcing of blocks, as highlighted in (Matsueda, 2009; Schiemann et al., 2017). These findings align with our observations made in this paper that the perturbations of the preexisting synoptic-scale eddies are prone to result in unstable and chaotic behavior in the evolution of blocking events. It is also a valuable topic to discuss the climatological seasonal impact of blocking, as mentioned in (Newman and Sardeshmukh,
550  1998).

*Code availability.* The codes that support the findings of this study are available from the corresponding author, Bin Shi, upon reasonable request.




*Data availability.* No data sets were used in this article.

## Appendix A: Derivation of the NMI model

In this section, we complement the details of the NMI model's derivation in Section 2.1.

### A1   The wave-superposition form of the preexisting synoptic-scale eddies streamfunction $\psi'_1$ and its phase velocities

Let us first put the asymptotic expansions of both the planetary-scale blocking anomaly and synoptic-scale eddies streamfunctions, (6a) and (6b), into the characteristic equation of synoptic-scale eddies (3b). Taking the lowest approximation, we obtain that $\psi'_1$ satisfies the $O(\epsilon^{\frac{3}{2}})$-order approximating equation of synoptic-scale eddies as

$$\left(\frac{\partial}{\partial t} + U\frac{\partial}{\partial x}\right)\left(\nabla^2\psi'_1 - F\psi'_1\right) + PV_y\frac{\partial\psi'_1}{\partial x} = 0. \tag{A1}$$

Then, according to the observation, we may assume that the synoptic-scale eddies streamfunction takes the following form as

$$\psi'_1(x,t;X_1) = \frac{2f_0(X_1)}{\epsilon^{\frac{3}{2}}}\left(\cos(k_1 x - \omega_1 t) - \cos(k_2 x - \omega_2 t)\right)\sin\left(\frac{my}{2} - \frac{\pi}{8}\right), \tag{A2}$$

where $f_0(X_1) = a_0\exp\left[-\mu(X_1 + \epsilon x_T)^2\right]$. Some simple substitution of variables tells us that the synoptic-scale streamfunction $\psi'_1$ takes the wave-superposition form of (7c). Taking the wave-superposition form (7c) into the $O(\epsilon^{\frac{3}{2}})$-order approximating

equation of synoptic-scale eddies (A1), the phase velocities in (8c) are derived.

### A2   The single-wave form of the blocking wavy anomaly stramfunction $\psi_1$ and its phase velocity

Putting the asymptotic expansions of both the planetary-scale blocking anomaly and synoptic-scale eddies streamfunctions, (6a) and (6b), into the characteristic equation of planetary-scale blocking anomaly (3a), we take the $O(\epsilon)$-order approximation and obtain that $\psi_1$ satisfies

$$\left(\frac{\partial}{\partial t} + U\frac{\partial}{\partial x}\right)\left(\nabla^2\psi_1 - F\psi_1\right) + PV_y\frac{\partial\psi_1}{\partial x} = 0. \tag{A3}$$

The blocking wavy anomaly streamfunction $\psi_1$ is assumed with the form (7a), where $B$ is the complex amplitude only dependent on the slow-varying variables, $X_1, X_2, \ldots$ and $T_1, T_2, \ldots$. Taking the single-wave form of the blocking wavy anomaly streamfunction (7a) into the $O(\epsilon)$-order approximating equation of the planetary-scale blocking anomaly streamfunction (3a), we derive that the phase velocity satisfies (8a).

### A3   The linear relationship of the complex blocking amplitude $B$ and the group velocity of the blocking wavy anomaly $\psi_1$

Putting the asymptotic expansions of both the planetary-scale blocking anomaly and synoptic-scale eddies streamfunctions, (6a) and (6b), into the characteristic equation of planetary-scale blocking anomaly (3a), we take the $O(\epsilon^2)$-order approximation and





obtain that $\psi_2$ satisfies


$$\left(\frac{\partial}{\partial x} + U\frac{\partial}{\partial x}\right)\left(\nabla^2\psi_2 - F\psi_2\right) + PV_y\frac{\partial\psi_2}{\partial x}$$

$$= -2\left(\frac{\partial}{\partial t} + U\frac{\partial}{\partial x}\right)\frac{\partial^2\psi_1}{\partial x\partial X_1} - \left(\frac{\partial}{\partial T_1} + U\frac{\partial}{\partial X_1}\right)\left(\nabla^2\psi_1 - F\psi_1\right) - PV_y\frac{\partial\psi_1}{\partial X_1} - J(\psi_1, \nabla^2\psi_1). \tag{A4}$$

Let the associated zonal-mean anomaly $\psi_2$ also satisfy the linear barotropic quasi-geostrophic equation. Then, we can obtain

$$2\left(\frac{\partial}{\partial t} + U\frac{\partial}{\partial x}\right)\frac{\partial^2\psi_1}{\partial x\partial X_1} + \left(\frac{\partial}{\partial T_1} + U\frac{\partial}{\partial X_1}\right)\left(\nabla^2\psi_1 - F\psi_1\right) + PV_y\frac{\partial\psi_1}{\partial X_1} - J(\psi_1, \nabla^2\psi_1) = 0.$$

Putting the single-wave form of the blocking wavy anomaly streamfunction (7a) into it, we obtain the group velocity (8) and

the linear relationship as

$$\frac{\partial B}{\partial T_1} + c_g\frac{\partial B}{\partial X_1} = 0. \tag{A5}$$

## A4 The static and sinusoidal form of the associated zonal-mean anomaly $\psi_2$

Putting the asymptotic expansions of both the planetary-scale blocking anomaly and synoptic-scale eddies streamfunctions, (6a) and (6b), into the characteristic equation of planetary-scale blocking anomaly (3a), we take the $O(\epsilon^3)$-order approximation and

obtain that $\psi_3$ satisfies

$$\left(\frac{\partial}{\partial t} + U\frac{\partial}{\partial x}\right)\left(\nabla^2\psi_3 - F\psi_3\right) + PV_y\frac{\partial\psi_3}{\partial x}$$

$$= -2\left(\frac{\partial}{\partial t} + U\frac{\partial}{\partial x}\right)\frac{\partial^2\psi_2}{\partial x\partial X_1} - \left(\frac{\partial}{\partial T_1} + U\frac{\partial}{\partial X_1}\right)\left(\nabla^2\psi_2 - F\psi_2\right) - PV_y\frac{\partial\psi_2}{\partial X_1}$$

$$\underbrace{-2\left(\frac{\partial}{\partial t} + U\frac{\partial}{\partial x}\right)\frac{\partial^2\psi_1}{\partial x\partial X_2}}_{\mathbf{I_1}} \underbrace{-\left(\frac{\partial}{\partial T_2} + U\frac{\partial}{\partial X_2}\right)\left(\nabla^2\psi_1 - F\psi_1\right)}_{\mathbf{I_2}} \underbrace{-PV_y\frac{\partial\psi_1}{\partial X_2}}_{\mathbf{I_3}}$$

$$\underbrace{-2\left(\frac{\partial}{\partial T_1} + U\frac{\partial}{\partial X_1}\right)\frac{\partial^2\psi_1}{\partial x\partial X_1}}_{\mathbf{II_1}} \underbrace{-\left(\frac{\partial}{\partial t} + U\frac{\partial}{\partial x}\right)\frac{\partial^2\psi_1}{\partial X_1^2}}_{\mathbf{II_2}}$$


$$\underbrace{-2J\left(\psi_1, \frac{\partial^2\psi_1}{\partial x\partial X_1}\right)}_{\mathbf{III_1}} \underbrace{-\left(\frac{\partial\psi_1}{\partial X_1}\frac{\partial\nabla^2\psi_1}{\partial y} - \frac{\partial\psi_1}{\partial y}\frac{\partial\nabla^2\psi_1}{\partial X_1}\right)}_{\mathbf{III_2}}$$

$$\underbrace{-J(\psi_1, \nabla^2\psi_2)}_{\mathbf{IV_1}} \underbrace{-J(\psi_2, \nabla^2\psi_1)}_{\mathbf{IV_2}}$$

$$\underbrace{-J(\psi_1', \nabla^2\psi_1')_P}_{\mathbf{V}} \tag{A6}$$

Let $\psi_3$ also satisfy the linear barotropic quasi-geostrophic equation. With the single-wave form of the blocking wavy anomaly streamfunction (7a), we know that $\nabla^2\psi_1$ is proportional to $\psi_1$. Hence, taking the zonal average of (A6), we obtain that the





associated zonal-mean anomaly streamfunction $\psi_2$ satisfies

$$\left(\frac{\partial}{\partial T_1} + U\frac{\partial}{\partial X_1}\right)\left(\frac{\partial^2\psi_2}{\partial y^2} - F\psi_2\right) + PV_y\frac{\partial\psi_2}{\partial X_1} = \frac{4mk^2}{\epsilon^2 L_y}\frac{\partial|B|^2}{\partial X_1}\cos(2my) \qquad (A7)$$

Obviously, the assumption of the associated zonal-mean anomaly with the form $\psi_2 = -\epsilon^{-2}g|B|^2\cos(2my)$ is reasonable, since

$$\left.\frac{\partial\psi_2}{\partial y}\right|_{y=0} = \left.\frac{\partial\psi_2}{\partial y}\right|_{y=L_y} = 0$$

satisfy the boundary condition, thus we obtain that the associated zonal-mean anomaly $\psi_1$ satisfies (7b). With the equality of zonal average (A7), the coefficient is obtained as

$$g = \frac{4mk^2(m^2 + k^2 + F)^2}{PV_y L_y\left[(4m^2 + F)(m^2 + F - k^2) - (m^2 + k^2 + F)^2\right]}.$$

### A5  Nonlinear time-evolution behavior of the complex amplitude $B$

Since the nondimensional blocking wavy anomaly streamfunction, $\psi_1$, is composed of the superposition of the single wave component, such as $\frac{1}{\epsilon}\sqrt{\frac{2}{L_y}}\sin\left(my - \frac{\pi}{4}\right)Be^{i(kx-\omega t)}$, and its conjugate $\frac{1}{\epsilon}\sqrt{\frac{2}{L_y}}\sin\left(my - \frac{\pi}{4}\right)\overline{B}e^{-i(kx-\omega t)}$, we only focus on the part $\frac{1}{\epsilon}\sqrt{\frac{2}{L_y}}\sin\left(my - \frac{\pi}{4}\right)Be^{i(kx-\omega t)}$ in our analysis for convenience. Then, we show the right-hand side of (A6) by five parts as

– **Part-I**: Putting $\psi_1 = \frac{1}{\epsilon}\sqrt{\frac{2}{L_y}}\sin\left(my - \frac{\pi}{4}\right)Be^{i(kx-\omega t)}$ into Part-I, we obtain that

$$\begin{aligned}
\mathbf{I}_1 + \mathbf{I}_2 + \mathbf{I}_3 =& 2\left(\frac{\partial}{\partial t} + U\frac{\partial}{\partial x}\right)\frac{\partial^2\psi_1}{\partial x\partial X_2} + \left(\frac{\partial}{\partial T_2} + U\frac{\partial}{\partial X_2}\right)\left(\nabla^2\psi_1 - F\psi_1\right) + PV_y\frac{\partial\psi_1}{\partial X_2} \\
=& -\frac{1}{\epsilon}\sqrt{\frac{2}{L_y}}\left(m^2 + k^2 + F\right)\left(\frac{\partial B}{\partial T_2} + c_g\frac{\partial B}{\partial X_2}\right)\sin\left(my - \frac{\pi}{4}\right)Be^{i(kx-\omega t)}.
\end{aligned}$$

– **Part-II**: With the phase velocity (8a) and the linear relationship (A5), we take $\psi_1 = \frac{1}{\epsilon}\sqrt{\frac{2}{L_y}}\sin\left(my - \frac{\pi}{4}\right)Be^{i(kx-\omega t)}$ into Part-II as

$$\begin{aligned}
\mathbf{II}_1 + \mathbf{II}_2 =& 2\left(\frac{\partial}{\partial T_1} + U\frac{\partial}{\partial X_1}\right)\frac{\partial^2\psi_1}{\partial x\partial X_1} + \left(\frac{\partial}{\partial t} + U\frac{\partial}{\partial x}\right)\frac{\partial^2\psi_1}{\partial X_1^2} \\
=& \frac{i}{\epsilon}\left[-\omega + kU + \frac{2kPV_y(m^2 - k^2 + F)}{(m^2 + k^2 + F)^2}\right]\frac{\partial^2\psi_1}{\partial X_1^2} \\
=& \frac{i}{\epsilon}\sqrt{\frac{2}{L_y}}\frac{kPV_y\left[3(m^2 + F) - k^2\right]}{(m^2 + k^2 + F)^2}\frac{\partial^2 B}{\partial X_1^2}\sin\left(my - \frac{\pi}{4}\right)Be^{i(kx-\omega t)}
\end{aligned}$$

– **Part-III**: With the property that $\nabla^2\psi_1$ is proportional to $\psi_1$, we known $\mathbf{III}_2 = 0$. With the property of Jacobians, we obtain $\mathbf{III}_1$ is proportional to $\cos(2my)$, thus

$$\mathbf{III}_1 + \mathbf{III}_2 \propto \cos(2my).$$





- **Part-IV**: Putting $\psi_1 = \frac{1}{\epsilon}\sqrt{\frac{2}{L_y}}\sin\left(my - \frac{\pi}{4}\right)Be^{i(kx-\omega t)}$ and $\psi_2 = -\epsilon^{-2}g|B|^2\cos(2my)$ into Part-IV, we obtain that

$$\mathbf{IV}_1 + \mathbf{IV}_2 = J(\psi_1, \nabla^2\psi_2) + J(\psi_2, \nabla^2\psi_1)$$


$$= -\frac{2i}{\epsilon^3}\sqrt{\frac{L_y}{2}}gkm(3m^2 - k^2)|B|^2B\sin\left(my - \frac{\pi}{4}\right)\sin(2my)e^{i(kx-\omega t)}$$

- **Part-V**: Here, we only consider the coefficient of the wave with wavenumber $k_2 - k_1$. Hence, we take the superposition form of $\psi_1'$ (A2) into Part-V and obtain that

$$\frac{1}{2L_x}\int_{-L_x}^{L_x}\mathbf{V}e^{-i(k_2-k_1)x}dx = \frac{if_0(X_1)}{\epsilon^3}\frac{m(k_1+k_2)(k_1^2-k_2^2)}{4}\cdot\frac{L_y}{2}e^{-i(\omega_2-\omega_1)t}$$

Taking some basic calculations, we obtain the following two equalities as


$$\int_0^{L_y}\sin^2\left(my - \frac{\pi}{4}\right)dy = \frac{L_y}{2}, \quad \text{and} \quad \int_0^{L_y}\sin^2\left(my - \frac{\pi}{4}\right)\sin(2my)dy = -\frac{L_y}{4}.$$

Filtering out the wave with the zonal wavenumber $k = 2k_0$ and the meridional wavenumber $m$ of the right-hand side of the $O(\epsilon^3)$-order expansion (A6), we obtain the forced NLS equation of the complex blocking amplitude $B$ with the periodic boundary condition (9).

**Appendix B: The wave-superposition form of the deformed synoptic-scale eddies streamfunction $\psi_2'$**

In the supplemental material, we complement the detailed derivation of the wave-superposition form of the deformed synoptic-scale eddies streamfunction $\psi_2'$ for the reference, which has been derived in (Luo, 2000, 2005; Luo et al., 2014, 2019). Putting the asymptotic expansions of both the planetary-scale and synoptic-scale streamfunctions, (2.6a) and (2.6b), into the characteristic equation of synoptic-scale eddies (2.3b), we obtain that $\psi_2'$ satisfies the $O(\epsilon^{\frac{5}{2}})$-order approximating equation of synoptic-scale eddies as


$$\left(\frac{\partial}{\partial x} + U\frac{\partial}{\partial x}\right)\left(\nabla^2\psi_2' - F\psi_2'\right) + PV_y\frac{\partial\psi_2'}{\partial x}$$

$$= -2\left(\frac{\partial}{\partial t} + U\frac{\partial}{\partial x}\right)\frac{\partial^2\psi_1'}{\partial x\partial X_1} - U\frac{\partial}{\partial X_1}\left(\nabla^2\psi_1' - F\psi_1'\right) - PV_y\frac{\partial\psi_1'}{\partial X_1} + \nabla^2\psi_S^{\star}$$

$$- J(\psi_1, \nabla^2\psi_1') - J(\psi_1', \nabla^2\psi_1). \tag{B1}$$

It is stated in (Luo, 2005) that the deformed synoptic-scale eddies streamfunction $\psi_2'$ is induced by the feedback of the blocking development and a modification to the preexisting synoptic-scale eddies, thus


$$-2\left(\frac{\partial}{\partial t} + U\frac{\partial}{\partial x}\right)\frac{\partial^2\psi_1'}{\partial x\partial X_1} - U\frac{\partial}{\partial X_1}\left(\nabla^2\psi_1' - F\psi_1'\right) - PV_y\frac{\partial\psi_1'}{\partial X_1} + \nabla^2\psi_S^{\star} = 0.$$





Hence, the $O(\epsilon^{\frac{5}{2}})$-order approximating equation (B1) is simplified as

$$\left(\frac{\partial}{\partial x} + U\frac{\partial}{\partial x}\right)\left(\nabla^2\psi'_2 - F\psi'_2\right) + PV_y\frac{\partial\psi'_2}{\partial x} = -J(\psi_1, \nabla^2\psi'_1) - J(\psi'_1, \nabla^2\psi_1). \tag{B2}$$

Since the preexisting synoptic-scale eddies $\psi'_1$ is the superposition of two singe waves, then we represent it as $\psi'_1 = \psi'_{1,k_1} + \psi'_{1,k_2}$, where the two single waves are respectively

$\quad \psi'_{1,k_1} = \dfrac{f(X_1)}{\epsilon^{\frac{3}{2}}}e^{i(k_1x - \omega_1 t)}\sin\left(\dfrac{my}{2} - \dfrac{\pi}{8}\right) + cc.$ and $\quad \psi'_{2,k_1} = \dfrac{2f(X_1)}{\epsilon^{\frac{3}{2}}}e^{i(k_2x - \omega_2 t)}\sin\left(\dfrac{my}{2} - \dfrac{\pi}{8}\right) + cc.$

Then, with the linearity of Jacobians, we separate the right-hand side of (B1) into two parts as

$$J(\psi_1, \nabla^2\psi'_1) + J(\psi'_1, \nabla^2\psi_1) = \underbrace{J(\psi_1, \nabla^2\psi'_{1,k_1}) + J(\psi'_{1,k_1}, \nabla^2\psi_1)}_{\mathbf{I}} + \underbrace{J(\psi_1, \nabla^2\psi'_{1,k_2}) + J(\psi'_{1,k_2}, \nabla^2\psi_1)}_{\mathbf{II}}.$$

– For the Part-**I**, with the property that the Laplacian of the streamfuncion is proportional to the streamfunction itself, we can obtain

$\quad \mathbf{I} = J(\psi_1, \nabla^2\psi'_{1,k_1}) + J(\psi'_{1,k_1}, \nabla^2\psi_1)$

$$= -\left(k_1^2 + \frac{m^2}{4}\right)J(\psi_1, \psi'_{1,k_1}) - \left(k^2 + m^2\right)J(\psi'_{1,k_1}, \psi_1)$$

$$= -\left(k_1^2 - k^2 - \frac{3m^2}{4}\right)\frac{f(X_1)}{\epsilon^{\frac{5}{2}}}\sqrt{\frac{2}{L_y}}\left[\underbrace{J\left(Be^{i(kx - \omega t)}\sin\left(my - \frac{\pi}{4}\right), e^{i(k_1x - \omega_1 t)}\sin\left(\frac{my}{2} - \frac{\pi}{8}\right)\right)}_{\mathbf{I}_1}\right.$$

$$\left. + \underbrace{J\left(\overline{B}e^{-i(kx - \omega t)}\sin\left(my - \frac{\pi}{4}\right), e^{i(k_1x - \omega_1 t)}\sin\left(\frac{my}{2} - \frac{\pi}{8}\right)\right)}_{\mathbf{I}_2} + cc. \right],$$

where $\mathbf{I}_1$ and $\mathbf{I}_2$ are calculated respectively as

$\quad \mathbf{I}_1 = J\left(Be^{i(kx - \omega t)}\sin\left(my - \frac{\pi}{4}\right), e^{i(k_1x - \omega_1 t)}\sin\left(\frac{my}{2} - \frac{\pi}{8}\right)\right)$

$$= imBe^{i[(k_1 + k)x - (\omega_1 + \omega)t]}\left[\left(\frac{k}{2} - k_1\right)\sin\left(\frac{3my}{2} - \frac{3\pi}{8}\right) + \left(\frac{k}{2} + k_1\right)\sin\left(\frac{my}{2} - \frac{\pi}{4}\right)\right]$$

and

$$\mathbf{I}_2 = J\left(\overline{B}e^{-i(kx - \omega t)}\sin\left(my - \frac{\pi}{4}\right), e^{i(k_1x - \omega_1 t)}\sin\left(\frac{my}{2} - \frac{\pi}{8}\right)\right)$$

$$= -im\overline{B}e^{i[(k_1 - k)x - (\omega_1 - \omega)t]}\left[\left(\frac{k}{2} + k_1\right)\sin\left(\frac{3my}{2} - \frac{3\pi}{8}\right) - \left(\frac{k}{2} - k_1\right)\sin\left(\frac{my}{2} - \frac{\pi}{4}\right)\right].$$





– For the Part-**II**, with the property that the Laplacian of the streamfuncion is proportional to the streamfunction itself, we can obtain

$$
\begin{aligned}
\mathbf{II} &= J(\psi_1, \nabla^2 \psi'_{1,k_2}) + J(\psi'_{1,k_2}, \nabla^2 \psi_1) \\
&= -\left(k_2^2 + \frac{m^2}{4}\right) J(\psi_1, \psi'_{1,k_2}) - \left(k^2 + m^2\right) J(\psi'_{1,k_2}, \psi_1) \\
&= -\left(k_2^2 - k^2 - \frac{3m^2}{4}\right) \frac{f(X_1)}{\epsilon^{\frac{5}{2}}} \sqrt{\frac{2}{L_y}} \left[ \underbrace{J\left(Be^{i(kx-\omega t)} \sin\left(my - \frac{\pi}{4}\right), e^{i(k_2 x - \omega_2 t)} \sin\left(\frac{my}{2} - \frac{\pi}{8}\right)\right)}_{\mathbf{II}_1} \right. \\
&\qquad\qquad\qquad \left. + \underbrace{J\left(\overline{B}e^{-i(kx-\omega t)} \sin\left(my - \frac{\pi}{4}\right), e^{i(k_2 x - \omega_2 t)} \sin\left(\frac{my}{2} - \frac{\pi}{8}\right)\right)}_{\mathbf{II}_2} + cc. \right],
\end{aligned}
$$

where $\mathbf{I}_1$ and $\mathbf{I}_2$ are calculated respectively as

$$
\begin{aligned}
\mathbf{II}_1 &= J\left(Be^{i(kx-\omega t)} \sin\left(my - \frac{\pi}{4}\right), e^{i(k_2 x - \omega_2 t)} \sin\left(\frac{my}{2} - \frac{\pi}{8}\right)\right) \\
&= imBe^{i[(k_2+k)x - (\omega_2+\omega)t]} \left[\left(\frac{k}{2} - k_2\right) \sin\left(\frac{3my}{2} - \frac{3\pi}{8}\right) + \left(\frac{k}{2} + k_2\right) \sin\left(\frac{my}{2} - \frac{\pi}{4}\right)\right]
\end{aligned}
$$

and


$$
\begin{aligned}
\mathbf{II}_2 &= J\left(\overline{B}e^{-i(kx-\omega t)} \sin\left(my - \frac{\pi}{4}\right), e^{i(k_2 x - \omega_2 t)} \sin\left(\frac{my}{2} - \frac{\pi}{8}\right)\right) \\
&= -im\overline{B}e^{i[(k_2-k)x - (\omega_2-\omega)t]} \left[\left(\frac{k}{2} + k_2\right) \sin\left(\frac{3my}{2} - \frac{3\pi}{8}\right) - \left(\frac{k}{2} - k_2\right) \sin\left(\frac{my}{2} - \frac{\pi}{4}\right)\right].
\end{aligned}
$$

Hence, we know that the deformed synoptic-scale eddies streamfunction $\psi'_2$ is the superposition of four kinds of waves,

$$
Be^{i[(k_j+k)x - (\omega_j+\omega)t]} \left[\left(\frac{k}{2} - k_j\right) \sin\left(\frac{3my}{2} - \frac{3\pi}{8}\right) + \left(\frac{k}{2} + k_j\right) \sin\left(\frac{my}{2} - \frac{\pi}{4}\right)\right]
$$

and

$\quad \overline{B}e^{i[(k_j-k)x - (\omega_j-\omega)t]} \left[\left(\frac{k}{2} + k_j\right) \sin\left(\frac{3my}{2} - \frac{3\pi}{8}\right) - \left(\frac{k}{2} - k_j\right) \sin\left(\frac{my}{2} - \frac{\pi}{4}\right)\right],$

where $k = 1, 2$. Taking the wave-superposition form of the deformed synoptic-scale eddies streamfunction $\psi'_2$ into (B1), we can obtain

$$
\begin{aligned}
\psi'_2 = &-\sum_{j=1}^{2} Q_j B \exp\left\{i\left[(k_j+k)x - (\omega_j+\omega)t\right]\right\} \left[p_j \sin\left(\frac{3my}{2} - \frac{3\pi}{8}\right) + r_j \sin\left(\frac{my}{2} - \frac{\pi}{8}\right)\right] \\
&+ \sum_{j=1}^{2} Q_j \overline{B} \exp\left\{i\left[(k_j-k)x - (\omega_j-\omega)t\right]\right\} \left[s_j \sin\left(\frac{3my}{2} - \frac{3\pi}{8}\right) + h_j \sin\left(\frac{my}{2} - \frac{\pi}{8}\right)\right]
\end{aligned}
\tag{B3}
$$



where the parameters are set for $j = 1, 2$ as

$$Q_j = -\frac{m}{4}\sqrt{\frac{2}{L_y}} f_0(x) \left( k_j^2 - k^2 - \frac{3m^2}{4} \right)$$

and

$$p_j = \frac{k - 2k_j}{PV_y \left\{ k_j + k - \left( \frac{k_j}{k_j^2 + \frac{m^2}{4} + F} + \frac{k}{k^2 + m^2 + F} \right) \left[ (k_j + k)^2 + \frac{9m^2}{4} + F \right] \right\}},$$

$$r_j = \frac{k + 2k_j}{PV_y \left\{ k_j + k - \left( \frac{k_j}{k_j^2 + \frac{m^2}{4} + F} + \frac{k}{k^2 + m^2 + F} \right) \left[ (k_j + k)^2 + \frac{m^2}{4} + F \right] \right\}},$$

$$s_j = \frac{k + 2k_j}{PV_y \left\{ k_j - k - \left( \frac{k_j}{k_j^2 + \frac{m^2}{4} + F} - \frac{k}{k^2 + m^2 + F} \right) \left[ (k_j - k)^2 + \frac{9m^2}{4} + F \right] \right\}},$$

$$h_j = \frac{k - 2k_j}{PV_y \left\{ k_j - k - \left( \frac{k_j}{k_j^2 + \frac{m^2}{4} + F} - \frac{k}{k^2 + m^2 + F} \right) \left[ (k_j - k)^2 + \frac{m^2}{4} + F \right] \right\}}.$$

*Author contributions.* Bin Shi constructed the basic idea of this paper, derived all formulas, and wrote the paper. Wenqi Zhang and Bin Shi coded the CNOP method in the NMI model and performed the experiments. All authors have read and approved the final manuscript.

*Competing interests.* The contact author has declared that none of the authors has any competing interests.

*Disclaimer.* Publisher's note: Copernicus Publications remains neutral with regard to jurisdictional claims in published maps and institutional affiliations.

*Acknowledgements.* This work was supported by Grant No.12241105 of NSFC and Grant No.YSBR-034 of CAS.





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
