# Peer review of "Optimal Disturbances of Blocking: A Barotropic View"

_EGUsphere, 2024_

## Referee Comment (RC2)

[revised manuscript text omitted]

[Figure]

[Figure]

Rex, D. F.: Blocking action in the middle troposphere and its effect upon regional climate, Tellus, 2, 275–301, 1950.

810 Rizzoli, P. M.: Planetary solitary waves in geophysical flows, in: Advances in Geophysics, vol. 24, pp. 147–224, Elsevier, 1982.

Schiemann, R., Demory, M.-E., Shaffrey, L. C., Strachan, J., Vidale, P. L., Mizielinski, M. S., Roberts, M. J., Matsueda, M., Wehner, M. F., and Jung, T.: The resolution sensitivity of Northern Hemisphere blocking in four 25-km atmospheric global circulation models, Journal of Climate, 30, 337–358, 2017.

Shi, B. and Ma, J.: The Sampling Method for Optimal Precursors of ENSO Events, arXiv preprint arXiv:2308.13830, 2023.

815 Shi, B. and Sun, G.: An adjoint-free algorithm for conditional nonlinear optimal perturbations (CNOPs) via sampling, Nonlinear Processes in Geophysics, 30, 263–276, 2023.

Shutts, G.: The propagation of eddies in diffluent jetstreams: Eddy vorticity forcing of 'blocking' flow fields, Quarterly Journal of the Royal Meteorological Society, 109, 737–761, 1983.

Sousa, P. M., Trigo, R. M., Barriopedro, D., Soares, P. M., and Santos, J. A.: European temperature responses to blocking and ridge regional patterns, Climate Dynamics, 50, 457–477, 2018.

820

Steinfeld, D. and Pfahl, S.: The role of latent heating in atmospheric blocking dynamics: a global climatology, Climate Dynamics, 53, 6159–6180, 2019.

Swanson, K. L.: Blocking as a local instability to zonally varying flows, Quarterly Journal of the Royal Meteorological Society, 127, 1341–1355, 2001.

825 Thorncroft, C. D., Hoskins, B. J., and McIntyre, M. E.: Two paradigms of baroclinic-wave life-cycle behaviour, Quarterly Journal of the Royal Meteorological Society, 119, 17–55, 1993.

Tibaldi, S. and Molteni, F.: On the operational predictability of blocking, Tellus A, 42, 343–365, 1990.

Tung, K. K. and Lindzen, R. S.: A theory of stationary long waves. Part I: A simple theory of blocking, Monthly Weather Review, 107, 714–734, 1979.

830 Vannitsem, S. and Nicolis, C.: Lyapunov vectors and error growth patterns in a T21L3 quasigeostrophic model, Journal of the atmospheric sciences, 54, 347–361, 1997.

Vautard, R.: Multiple weather regimes over the North Atlantic: Analysis of precursors and successors, Monthly weather review, 118, 2056–2081, 1990.

Wang, Q. and Mu, M.: A new application of conditional nonlinear optimal perturbation approach to boundary condition uncertainty, Journal of Geophysical Research: Oceans, 120, 7979–7996, 2015.

835

Weijenborg, C., de Vries, H., and Haarsma, R. J.: On the direction of Rossby wave breaking in blocking, Climate dynamics, 39, 2823–2831, 2012.

Witte, J. C., Douglass, A. R., Da Silva, A., Torres, O., Levy, R., and Duncan, B. N.: NASA A-Train and Terra observations of the 2010 Russian wildfires, Atmospheric Chemistry and Physics, 11, 9287–9301, 2011.

840 Woollings, T., Barriopedro, D., Methven, J., Son, S.-W., Martius, O., Harvey, B., Sillmann, J., Lupo, A. R., and Seneviratne, S.: Blocking and its response to climate change, Current climate change reports, 4, 287–300, 2018.

Zabusky, N. J. and Porter, M. A.: Soliton, Scholarpedia, 5, 2068, https://doi.org/10.4249/scholarpedia.2068, revision #186585, 2010.

Zhang, F., Sun, Y. Q., Magnusson, L., Buizza, R., Lin, S.-J., Chen, J.-H., and Emanuel, K.: What is the predictability limit of midlatitude weather?, Journal of the Atmospheric Sciences, 76, 1077–1091, 2019.

---

## Author Comment (AC1)

**Response to Reviewers' Comments**
**Optimal Disturbances of Atmospheric Blocking: A Barotropic View**

Bin Shi[*] [1,2], Dehai Luo[3,4], and Wenqi Zhang[3]

[1]*Shanghai Institute for Mathematics and Interdisciplinary Sciences, Shanghai 200433, China*
[2]*Research Center for Mathematics and Interdisciplinary Sciences, Fudan University, Shanghai 200433, China*
[3]*Institute of Atmospheric Physics, Chinese Academy of Sciences, Beijing 100029, China*
[4]*College of Earth and Planetary Sciences, University of Chinese Academy of Sciences, Beijing 100049, China*

November 22, 2024

We thank the associate editor and the reviewers for a thorough reading of the manuscript and for their helpful comments. We appreciate the effort the reviewers put in and the very constructive feedback he or she provided.

**1 The First Reviewer's Comments**

1. *In several previous studies, especially in Jiang and Wang [2010] and Jiang et al. [2011], the CNOP has been used to investigate the onset of blocking, which is very similar to the present study. But these papers were not cited. I suggest that the authors should discuss and clarify the differences between the work of Jiang and Wang [2010] and Jiang et al. [2011] and this manuscript.*

   Thanks for the helpful comments.We have incorporated a discussion of the suggested references, Jiang and Wang [2010] and Jiang et al. [2011], in Line 95 – Line 99: " **Moreover, CNOP has been specifically applied to studies of blocking onset; Mu and Jiang [2008] utilized a 3-layer T21L3 quasi-geostrophic (QG) model with CNOP to analyze blocking onset mechanisms, focusing on the blocking index and squared norm of the streamfunction. Their findings identified two types of CNOPs, one that amplifies and one that diminishes the blocking, with further applications to strong zonal flow [Jiang et al., 2011] and climatological flow anomalies over the Atlantic and Pacific Oceans [Jiang and Wang, 2010].**" Additionally, in Line 102 – Line 103, we have added: "**In contrast, this paper employs the NMI model developed by Luo [2000, 2005], Luo et al. [2014, 2019] to investigate...**"
* * *
[*]Corresponding author: shibin@lsec.cc.ac.cn

2. *Lines 221 and 234, $J(b0; B0, F0, U)$ should be $J(b_0; B_0, F_0, U)$.*

   Thanks, the two typos have been revised.

3. *In this manuscript, the simulation results of the model are not validated. Could you add a subsection to check whether the model can well reproduce the basic feature of blockings?*

   Thanks for pointing this out. The basic feature of blockings has indeed been well reproduced in Luo et al. [2019]. Following your suggestion, we have added detailed sentences in detail in Line 192 – Line 195 to emphasize this point: "**Building on this, the potential vorticity theory within the NMI model effectively captures the fundamental features of atmospheric blockings, as demonstrated in Luo et al. [2019]. In that study, Figure 1 presents observational data, while Figure 3, 4, 8 and 11 reproduce essential features of blockings.** "

4. *In some previous studies about CNOP, two types of CNOP can be obtained. But in this study, only one CNOP is calculated. Whether there are two types of CNOP: one strengthens the blocking and the other weakens the blocking?*

   Thank you for the insightful comments. In response, we explore the CNOP of the NMI model, which is governed solely by the NLS equation and uses the squared norm of the blocking amplitude as the objective function, a physically meaningful quantity [Colliander et al., 2010]. In the revised version, we added some sentences to explain: "**It is important to note that there is a key distinction from detecting blocking onset in the T21L3 QG model [Mu and Jiang, 2008, Jiang and Wang, 2010, Jiang et al., 2011]: the NMI model involves an NLS equation focused only on the blocking amplitude, with the objective function being the squared norm of the blocking amplitude, a physically meanfully quantity [Colliander et al., 2010]. In contrast, the QG model is related to the streamfunction, allowing the objective function to be either the blocking index or squared norm of the streamfunction [Mu and Jiang, 2008, Jiang and Wang, 2010, Jiang et al., 2011].**"

5. *From Figures 3a and 3b, we can see that the error decays in the first five days. Why? What is the physical mechanism? In Lines 316-318, the description may be inaccurate. We can clearly observe that the error decays, rather than that the growth is weaker.*

   Thanks for pointing this out. We revised the sentences in Line 326 – Line 331 to read: "**In Figures 3a, we observe that, during the blocking growth phase, the optimal disturbance exhibits minimal or even negative growth, probably due to the suppression of perturbation growth by the increasing blocking amplitude. In Figures 3b, this effect becomes more pronounced as the rapid growth of the blocking amplitude further inhibits and even reverses the growth of the perturbation. During other periods, however, the growth rate of the perturbation accelerates while the blocking amplitude grows more slowly, further supporting this interpretation.**"

6. *In this manuscript, the initial perturbation and the perturbation of the preexisting synoptic-scale eddies are both considered. This is remarkably different from previous studies by Jiang and Wang [2010] and Jiang et al. [2011], in which only initial perturbation was taken into account. In the realistic situation, these two types of perturbations may simultaneously exist. Could you give a discussion about the effects of the two types of perturbations?*

Thanks for the helpful comments. When the two types of perturbations simultaneously exist, the optimal distribution of the preexisting synoptic-scale eddies dominates. In the revised version, we have added a new subsection 4.4 with a title **Comparison of the effects of two types of optimal disturbaces** to discuss the two types of perturbations in detail in Line 465 – Line 480: "**In this scenario, we examine the effects of two types of optimal disturbances and the impact that arises when both coexist simultaneously. Referring to the nonlinear growth of optimal disturbances for the initial blocking amplitude in Figure 3 and for the preexisting synoptic-scale eddies in Figure 7, we present their individual effects, along with the nonlinear growth of their combined optimal disturbance under the constraint $\gamma = 1$ in Figure 10. When these two optimal disturbances are combined, the optimal disturbance associated with the preexisting synoptic-scale eddies becomes dominant. In comparison, the contribution from the optimal disturbance of the initial blocking amplitude is nearly negligible. This dominance is further confirmed in both the nonlinear growth (Figure 10a) and the nonlinear relative growth (Figure 10b).**

[Figure]

[Figure]

(a) (standard) Error Growth               (b) Relative Error Growth

Figure 1: Nonlinear growth of two types of optimal disturbances and their combined disturbance. The optimal disturbances are derived using (14) and (16) with the constraint $\gamma = 1$.

**To further validate the dominance of the optimal disturbance from preexisting synoptic-scale eddies as seen in Figure 10, we present the nonlinear evolution of the instantaneous total stream function field $\psi_T$ in Figure 11. Here, we observe that, although the initial stream function patterns align with the optimal disturbance for the initial blocking amplitude, over time, the disturbance from the preexisting synoptic-scale eddies becomes dominant. The cases with and without the initial blocking amplitude become nearly identical, with only intensified**

eddy straining and wave breaking effects. Consequently, the impact of the initial blocking amplitude disturbance is negligible compared to that of the preexisting synoptic-scale eddies. ”

[Figure]

(a) Initial Blocking Amplitude

(b) Preexisting Synoptic-Scale Eddies

(c) Combined Optimal Disturbances

Figure 2: Nonlinear evolution of the instantaneous total stream function field $\psi_T$, with background fields added by **(a)** the optimal disturbance for the initial blocking amplitude, **(b)** optimal disturbance for the preexisting synoptic-scale eddies, and **(c)** the combined optimal disturbances.

7. *In figure 8, more wave breakings are caused by the perturbation of preexisting synoptic-scale eddies. What is the physical reason?*

Thanks for pointing this out. According to the NMI model, wave-breaking phenomena are

driven by preexisting synoptic-scale eddies [Luo et al., 2019]. In this work, we confirm in Section 4 that atmospheric blocking is highly sensitive to the optimal disturbance originating from these eddies. Since eddy straining and wave breaking are key features of atmospheric blocking, the optimal disturbance of the preexisting synoptic-scale eddies results in more pronounced wave-breaking events.

8. *Please confirm $\gamma = 2$ in figures 10c and 11c.*

   Thanks for pointing this out. We have corrected $\gamma = 2$ to $\gamma = 1$ in figures 10c and 11c.

9. *The font size is too small in figures 4 and 8.*

   Thank you for bringing this to our attention. In the Copernicus-LaTeX package, figure captions are formatted with a smaller font size than the main text throughout the paper. The font size in Figures 4 and 8 is consistent with other figures and adheres to the package's formatting guidelines.

**2 The Second Reviewer's Comments**

1. *Line 1: What makes it "optimal". I think this should be very briefly made clear to the nonspecialist reader already at this point.*
   Thanks for pointing this out. We have added Footnote 1 after this sentence to clarify the term "optimal" as follows: **These "optimal disturbances" are configurations that maximize the intensity of a blocking event over time.**

2. *Line 3: Why "incrementally"? Do we need this word here at all?*
   Thanks, we have deleted the word "incrementally".

3. *Line 5: What would be such a thing, for example? Btw, I suppose, this is the attempt of explaining what is meant earlier, and in the title, by "optimal". Also, are you sure "influence" is the right word here?*
   Thank you for pointing this out. What we aim to convey is that during the evolution, the optimal disturbance of the initial blocking amplitude serves only to intensify the blocking amplitude, while other features of the atmospheric blocking remain consistent with the unperturbed motion of the atmospheric blocking. We have revised the sentence to replace **influence** with **effect** as "**Over time, this disturbance primarily intensifies the blocking amplitude itself, with little effect on surrounding atmospheric features.**"

4. *Line 8 — Line 9: This is a very obscure statement. I wonder if it would be clear to an expert on dynamic meteorology.*
   Thanks for pointing this out. We have revised the sentence for clarity as follows: " **However, the nonlinear evolution of this disturbance diverges markedly from that of disturbances in the initial blocking amplitude. It not only strongly intensifies the amplitude of blockings but also alters their structure, rendering eddy straining and wave breaking more chaotic and predominant.** "

5. *Line 9: Does it not contradict what you wrote on line 5?*
   This does not contradict what we wrote on line 5. Line 5 pertains to the optimal disturbance in the initial blocking amplitude, while the statement here refers to the optimal disturbance in preexisting synoptic-scale eddies.

6. *Line 13: Extremes are not necessarily less predictable [Bódai, 2015]. Is this just an untested hypothesis here, or you have reference for it?*
   Thanks for pointing the reference [Bódai, 2015] out and noting that extremes are not necessarily less predictable. We have revised the sentence to: "**Such eddy perturbations are more likely to trigger weather extremes, such as extreme cold events in North America and Eurasia [Yao et al., 2023], sometimes challenging predictability.**" This adjustment aligns with the discussion in [Yao et al., 2023] about how certain weather extremes can increase prediction challenges.

7. *Line 25: Please rephrase this. Make it sounds like we already know a lot about blocking, as is the case. Just a remark: Predicting the future is arguably the greatest power. But understanding a certain aspect of the phenomenon might not help us obtain better predictability. I'm not saying that such understanding has no value. Only that they should not be "marketed" as "predictive understanding".*
   Thanks for pointing this out. We have revised the sentence as "Therefore, **understanding the mechanisms behind blocking, a phenomenon already extensively studied, is essential for advancing our knowledge of these high-impact weather events and their role in atmospheric dynamics, even if such insights do not always translate directly into improved predictability (Kautz et al., 2022).**"

8. In Line 29, we have deleted the words "**the mechanism study of**".

9. In line 31, we have deleted the word "**by**" and have added the word "**that**".

10. In line 63, we have revised the word "**interply**" as "**interplay**".

11. In line 83, we have revised the typos "**block**" as "**blockinng**" and "**midlatitudes**" as "**midlatitude**", and added "**the**" before "**midlatitude**".

12. *In line 85, What are these?*
    To explain this, we have added Footnote 2 after this sentence as "**The normal and nonnormal modes are specifically defined in terms of linear instability. A normal mode refers to a linear operator that commutes with its transpose, i.e., $LL^\top = L^\top L$, while a nonnormal mode satisfies $LL^\top \neq L^\top L$.**".

13. In line 90, we have added "**function**" before "**value**".

14. In line 91, we have deleted "**or says**" and in line 92, we have deleted "**being studied**".

15. *In line 98, would be better to name them here.*
    Thanks. We have named them there and revised the sentence as "**More recently, advancements in machine learning have further improved the efficiency and application of CNOP, particularly in the Lorenz-96 model and the Zebiak-Cane model for the El Niño-Southern Oscillation events (Shi and Sun, 2023; Shi and Ma, 2023).**"

16. In line 102, we have revised the sentence as "**Section 2 outlines the derivation of the NMI model and the basic CNOP settings used to determine optimal disturbances.**"

17. *In line 114, please develop the language a bot here so that it makes a more precise and clear statement.*
    Thanks, we have revised the sentence as "**By maximizing these objective functions, we aim to identify the optimal perturbations that lead to the maximum total amplitude of blockings.** ".

18. *In line 117, how can a "parameter" have a gradient?*
    In the classical book [Cushman-Roisin and Beckers, 2011, Section 2.3], $f$ is referred to as the Coriolis parameter. The meridional gradient of the Coriolis parameter is the derivative of $f$ with respect to $y$.

19. *In line 117 — line 118, the physical dimension of $\beta$ is $rad * m$, if $[\beta_0] = rad/s$, which does not seem nondimensional.*
    The dimension of $\beta_0 = \frac{\partial f}{\partial y}$ is given by $[f]/L = m^{-1}s^{-1}$. Thus the dimension of $\beta$ can be expressed as

    $$\beta \sim \frac{1}{ms} \cdot \frac{m^2}{\frac{m}{s}} = 1$$

    Hence, $\beta$ is the nondimensional parameter.

20. *In line 118 — line 120, please use a mathematical symbol for the dimensional quantity too, e.g. u, so you can write $u = 7m/s$.*
    Thanks, we have replaced "**diemension**" by "**scale**' and 'revised the sentence as "Typically, the background westerly wind is observed to have a speed of approximately $\boldsymbol{u = 7\ m/s}$ (Luo, 2005). Considering that the **scale** of wind speed is $\boldsymbol{[u] = 10\ m/s}$, we set the nondimensional wind speed as $\boldsymbol{U = u/[u] = 0.7}$"

21. In line 121, again, poor English and poor notation. Thanks, we have revised the sentence as "**In this context, the zonal westerly wind varies only in the meridional direction and is expressed as $U = U(y)$.** "

22. In line 122, I'm sure this is not used like this in English. I don't know why "lateral" refers to a boundary condition to do with the meridional direction. Otherwise, i would write, "... boundary conditions that are periodic with respect to the zonal direction $x$". Thanks for pointing this out. We have revised the sentence as "**The boundary conditions are periodic in the zonal direction $x$ and of Dirichlet type in the meridional direction $y$, with specified mean values.** "

23. Comments in Table 1

    (1) *We are not informed of the dimensional width of the beta channel.*
        Thanks, the dimensional width of the beta channel is $\boldsymbol{L_y = 5L}$. Hence, $L_y = 5$ is the nondimensional width.

    (2) *wave numbers cannot be dimensional anyway*
        Thanks, we have deleted all the "**nondimensional**" before zonal number.

(3) *Please have a clearance between the bottom line of the table and the figure caption. I.e. make sure that you leave a bottom margin.*

Thanks, we have adjusted the position of the caption by moving it further down to ensure clarity and to create a clear margin between the bottom line of the table and the figure caption.

24. *In line 125, $L_x$ was not defined in Table 1. or anywhere. I also don't know what the function J() and F are.*

Thanks for the helpful comment. We have clarified the definition of $L_x$ in Table 1, specifying it as the zonal boundary, a nondiemnsional parameter with $L_x = 5.75$. Additionally, we included the definition of $J(\cdot, \cdot)$, identifying it as the Jacobian determinant. The definition of $F$, the Froud number, was already provided at the beginning of Section 2.1.

25. *In line 130, I suspect it has a more proper English name.*

Thanks, we have revised "**basic**" as "**background**".

26. *In line 131, okay, but what is its mathematical definition?*

The mathematical definition is given by the following sentence, that is,

$$\psi \propto e^{2k_0 x - \omega t}.$$

27. *In line 131, why do you call it "preexisting" when it is a function of time?*

Yes, $\psi'(x, y, t)$ represents the synoptic-scale eddies that evolve over time. However, the focus here is on the initial condition, $\psi'(x, y, 0)$. Perturbations in this initial condition can lead to significant changes, which form the core of this study and are detailed in the following section. Due to the critical role of the initial condition, we refer to it as the preexisting synoptic-scale eddies, which subsequently evolve over time.

28. *In line 133, does this arise because of topography, or would you have it even on an aquaplanet?*

Thanks, we revised the expression as "**a wave with a single wavenumber $k = 2k_0$** ".

29. *In line 135, "single" in what sense? I suspect you can simply delete this word.*

Thanks, we have deleted the word "**single**".

30. *In line 136, we are not told here anything.*

Yes, both $\omega_1$ and $\omega_2$ are unknown here, and they are derived in equation (8c).

31. *In line 153,I don't know what are $X_k$ and $T_k$.*

To clarify, here is the multiscale decomposition: $X_k = \epsilon^k x$ $T_k = \epsilon^k t$, which represent the $\epsilon^k$-scale spatial variable and temporal variable, respectively.

32. In line 168, the word "**wavy**" is used to emphasize that the blocking anomaly is $\psi_1$ is wavy, which is then compared with the associated zonal-mean anomaly $\psi_2$.

33. *In line 173, it's not great to have a footnote for an equation. I would add it in a bracket, starting with (Note that....)*

Thanks, we have changed the footnote as a bracket after the equation as "(**Noting that the external force $F_0$, acting as a filter for the waves, indeed serves as the core ingredient of the preexisting synoptic-scale eddies $\psi_1'$. Therefore, unless specifically**

mentioned, we use the external force $F_0$ to represent the preexisting synoptic-scale eddies.)"

34. *In line 174, and what is k?*
Thanks, $k = 2k_0$ is the zonal wavenumber of the planetary-scale blocking anomaly, which is shown in Table 1.

35. *In line 222, I don't really understand this. Is it integrated over some time? But what is the time horizon then?*
No, the calculation is not integrated over time. The time horizon is fixed at $T$. In equation (13), we fix the time horizon $T$, the external force $F_0$, and the speed of the background westerly wind $U$. For different initial blocking amplitudes, $B_0$ and $B_0 + b_0$, we compute the resulting blocking amplitudes at $T$, denoted as $B(T; B_0 + b_0, F_0, U)$ and $B(T; B_0, F_0, U)$, respectively. The objective function in equation (13) is then expressed as:

$$J(b_0; B_0, F_0, U) = \|B(T; B_0 + b_0, F_0, U) - B(T; B_0, F_0, U)\|^2.$$

In equation (14), we determine the maximum value of the objective function $J(b_0; B_0, F_0, U)$ within the domain $\{b_0 : \|b_0\| \le \rho\}$.

36. In line 223 — line 224, we have revised the sentence as "**By abbreviating** $J(b_0; B_0, F_0, U)$ **as** $J(b_0)$**, we simplify the notation, making it more convenient for subsequent discussions and calculations. This concise form allows for easier reference to** $J(b_0)$ **without any loss of generality.** "

37. In the caption of Figure 1, the term "**nondimensionalization**" in the brackets clarifies that the variable is dimensionless. Following your suggestion, we have removed the word "**nondimensionalization**" in the revised version. We also remove the word "**nondimensionalization**" in the caption of Figure 6.

38. In line 314. we have revised the phrase as "**transitioning from the same initial unit of one to different ratios**".

39. In line 332, we have revised the clause as "**when the optimal disturbances are added to the initial blocking amplitude**".

40. *In line 337, isn't this expected? What do we gain in addition by using this methodology?*
Yes, the description of the phenomena of eddy straining and wave breaking becoming more prominent is intended to contrast with the effects caused by the optimal disturbance of the preexisting synoptic-scale eddies, as discussed in Section 4.2 and illustrated in Figure 8.

41. *In line 351 — line 352, you have just said this in the first sentence. Can you not decide which one is more beautiful? Please provide a range in units of days*
Thanks for pointing this out. We have revised the sentence as " **Nevertheless, as noted by Kautz et al. (2022), accurately predicting atmospheric blocking on the medium-range weather timescale ($\sim 10$ days) remains a challenge.**".

42. *In line 355 — line 356, I'm sure you can say it shorter.*
Thanks, we have shortened the sentence to "**This approach seems suitable for using the NMI model to test if the optimal disturbance causes larger forecast errors.** ".

43. *In line 361 — line 364, rephrase it with good English*
    Thanks, we have rephrased the sentences as "**With the time interval $T$ and the size parameter $\gamma = 1$ fixed, the optimal disturbance at a later stage yields a greater error, as shown by the ratios:** $1.8776$ **for nonlinear growth and** $1.8781$ **for relative nonlinear growth.** "

44. *In line 395, do you say this because in a linear framework of analysis, the picture would be qualitatively different? If so, can you show that or reference a paper where it shows?*
    No, the qualitative difference lies in the nonlinear growth behaviors of two distinct scenarios. In Figures 2a and 3a, the nonlinear growth of the optimal disturbance of the initial blocking amplitude is relatively gradual as the parameter size $\gamma$ increases incrementally from 0.25 to 1. In contrast, Figure 7a illustrates the nonlinear growth of the optimal disturbance of the preexisting synoptic-scale eddies, which is significantly sharper under the same incremental increase in $\gamma$.

45. *In Figure 6, have a bit more spacing between the lines of the legend so the square brackets do not touch.*
    Thanks for pointing this out. We have adjusted the spacing between the lines in the legend of Figure 6 to ensure the square brackets do not touch. And also we have deleted the word "**nondimensionalization**" in the caption.

46. *In line 407, why do you say "potential"? Do you mean potentially in reality because what you see here is in a model and we are not sure how well it represents reality in this specific regard?*
    Yes, the use of "potential" is deliberate because the results presented here are derived from a model. While the model provides valuable insights, its ability to fully capture and represent reality in this specific context remains uncertain. Therefore, "potential" reflects the cautious interpretation of the findings.

47. *In line 408, wouldn't it be better to be more cautious and say "possible" instead of "probable" (or "likely")? Please go ahead with "likely" if you have to confidence. I don't.*
    Thanks, we have used the word "**possible**" instead of "**probable**".

48. *In line 420 — line 421, consider if this makes sense. What is added to what by what? I'm not even sure that the "* to * by *" structure is good.*
    Thanks for pointing this out. We have revised the clause as "when the optimal disturbance is added to the preexisting synoptic-scale eddies."

49. In line 421, Figure 8a is the **same** as Figure 4a, as both depict the evolution of the instantaneous total streamfunction without any perturbations.

50. *In line 426, how do we see that this is actually chaotic – as in the theory of dynamical systems? I'm not saying i don't believe it, but just now i don't know how to verify this.*
    Here, the word "**chaotic**" refers to a more disordered and unstable evolution of the system. A comparison of Figure 8 and Figure 4 shows that when the optimal disturbance is added to the preexisting synoptic-scale eddies, the evolution of the blocking becomes more disordered and unstable, in contrast to when the optimal disturbance is added to the initial blocking amplitude.

51. In line 435 — line 437, we have revised the sentences as "**Specifically, we investigate whether larger forecast errors occur at a later stage by comparing the optimal disturbances over different time periods, such as the early stage ( 0 to 10 days) and the later stage (5 to 15 days) of the evolution of the blocking.**"

52. In line 439, we have used the word "**is shifted**" instead of "**offsets**".

53. In line 442, we have used the word "**fast**" instead of "**more predominantly**".

54. *In line 444, the predictability of blocking to take place (soon), or just predictibaility of the field when blocking takes place?*
   Thanks for pointing this out. We have revised the part as " **reduces** the predictability of **the field when the blocking event takes place** "

55. In line 459, we have revised the rephase "**with a decrement of**" as "**in increments of**".

56. Comments in Figure 10.

   (1) *Do you understand why the peak shifts with decreasing U? Is there a simple way to see this? I'm looking for a physical explanation. But don't worry if there isn't a simple explanation.*
   The potential vorticity theory, as presented in [Luo et al., 2019], established that the coefficient of the dispersive term is proportional to $PV_y$, i.e., $\lambda \propto PV_y = \beta + FU$, while the coefficient of the nonlinear term is inversely proportional to $PV_y$, i.e., $\delta \propto 1/PV_y = 1/(\beta + FU)$. Hence, as the background westerly wind $U$ decreases, the coefficient of the dispersive term $\lambda$ also decreases, resulting in a more concentrated spatial pattern under weaker wind conditions. MFurthermore, as the background westerly wind increases, the peaks shift eastward, reflecting a form of equilibrium.

   (2) *Why does the error grow faster with smaller background wind? I'm embarased but I thought it would be the opposite.*
   The potential vorticity theory, as presented in [Luo et al., 2019], established that the coefficient of the dispersive term is proportional to $PV_y$, i.e., $\lambda \propto PV_y = \beta + FU$, while the coefficient of the nonlinear term is inversely proportional to $PV_y$, i.e., $\delta \propto 1/PV_y = 1/(\beta + FU)$. Hence, as the background westerly wind $U$ decreases, the coefficient of the nonlinear term $\delta$ increases, leading to faster error growth under weaker background wind conditions.

57. *In line 474 — line 475, hmm, does this mean that you are not showing the linear part of the growth e.g. in Fig. 10 b, ONLY the nonlinear part?*
   The error growth analyzed in the NMI model, governed by the forced nonlinear Schrödinger equation, is entirely nonlinear. In this analysis, we do not account for error growth arising solely from the linear component.

58. In line 482, we have revised the rephase "**with a decrement of**" as "**in increments of**".

59. *In line 494, is this the new thing in this paper? I mean, did you study nonlinear growth before but doing the calculation in some other way? If so, what is the difference that you observe, and did you figure out the reason?*

Yes, this paper first uses the CNOP approach (nonlinear optimization) to investigate the nonlinear error growth of the NMI model. Previous methods for the analysis of the error growth cannot be used to investigate the nonlinear partial differential equations, only for the linear partial differential equations.

60. *In line 500, is this the new thing in the paper? If so, did you arrive at it because 1. you used a new model or 2. new definition of the growth or 3. a new way of calculating the growth that was previously defined?*
Yes, the NMI model we use is novel. Additionally, we employ the innovative CNOP approach to uncover a remarkable phenomenon related to the optimal disturbance of preexisting synoptic-scale eddies, including eddy straining and wave breaking.

61. *In line 508 — line 509, if you say "probable", shouldn't you say "would" instead of "can". That is, shouldn't we have an agreement bw the level of confidence of the statements?Perhaps now, as i referred to my paper that extremes might not be less predictable depending perhaps on how you define predictability.*
Thanks for the suggested reference [Bódai, 2015]. We have deleted the sentence "**The perturbations of these eddies may be a probable cause of weather extremes and can reduce predictability.**".

62. In line 531, we have revised the sentence as " **Exploring how various types of perturbations in these models affect error growth would be highly intriguing.** "

63. *In line 546, is this really a term in use regarding atmospheric blocking?*
Thanks for pointing this out. We have revised the word "**blocks**" as "**blockings**".

64. In **code availability**, we have deleted the word "**reasonable**".

65. In line 555, we have revised the phrase "**In this section**" as "**Here**".

66. In line 556 and all in the appendix, we have revised the word "**put**" as "**substitute**".

67. In line 604 — line 605, we have revised the phrase "**With the equality of zonal average (A7)**" as "**Using the zonal average equation (A7)**".

68. In line 620 — line 621, we have revised the sentence as "**Using the property that $\nabla^2\psi_1$ is proportional to $\psi_1$, we deduce that III$_2$ = 0. Furthermore, leveraging the properties of Jacobians, we find that III$_1$ is proportional to** $\cos(2my)$**. Thus:**"

69. In line 629, we have revised the phrase "**Taking some basic calculations**" as "**Through straightforward calculations**".

70. In line 635, we have revised the word as "**Appendix B**".

71. In Appendix B, we have changed the color of all the equations from red to black.

72. In line 648, we have added "**the streamfunction of**" before the preexisting synoptic-scale eddies.

73. In line 681, we have revised the word "**taking**" as "**substituting**".

**References**

T. Bódai. Predictability of threshold exceedances in dynamical systems. *Physica D: Nonlinear Phenomena*, 313:37–50, 2015.

J. Colliander, M. Keel, G. Staffilani, H. Takaoka, and T. Tao. Transfer of energy to high frequencies in the cubic defocusing nonlinear schrödinger equation. *Inventiones mathematicae*, 181(1):39–113, 2010.

B. Cushman-Roisin and J.-M. Beckers. *Introduction to geophysical fluid dynamics: physical and numerical aspects*. Academic press, 2011.

Z. Jiang and D. Wang. A study on precursors to blocking anomalies in climatological flows by using conditional nonlinear optimal perturbations. *Quarterly Journal of the Royal Meteorological Society*, 136(650):1170–1180, 2010.

Z. Jiang, M. Mu, and D. Wang. Optimal perturbations triggering weather regime transitions: Onset of blocking and strong zonal flow. *Advances in Atmospheric Sciences*, 28:59–68, 2011.

D. Luo. Planetary-scale baroclinic envelope Rossby solitons in a two-layer model and their interaction with synoptic-scale eddies. *Dynamics of atmospheres and oceans*, 32(1):27–74, 2000.

D. Luo. A barotropic envelope Rossby soliton model for block-eddy interaction. Part I: Effect of topography. *Journal of the Atmospheric Sciences*, 62(1):5–21, 2005.

D. Luo, J. Cha, L. Zhong, and A. Dai. A nonlinear multiscale interaction model for atmospheric blocking: The eddy-blocking matching mechanism. *Quarterly Journal of the Royal Meteorological Society*, 140(683):1785–1808, 2014.

D. Luo, W. Zhang, L. Zhong, and A. Dai. A nonlinear theory of atmospheric blocking: A potential vorticity gradient view. *Journal of the Atmospheric Sciences*, 76(8):2399–2427, 2019.

M. Mu and Z. Jiang. A method to find perturbations that trigger blocking onset: Conditional nonlinear optimal perturbations. *Journal of the atmospheric sciences*, 65(12):3935–3946, 2008.

Y. Yao, W. Zhuo, Z. Gong, B. Luo, D. Luo, F. Zheng, L. Zhong, F. Huang, S. Ma, C. Zhu, and T. Zhou. Extreme cold events in north america and eurasia in november-december 2022: a potential vorticity gradient perspective. *Advances in Atmospheric Sciences*, 40:953–962, 2023.